# Pathogen evasion of social immunity

Miriam Stock[1,4], Barbara Milutinović ◉[1,2,4] ✉, Michaela Hoenigsberger ◉[1], Anna V. Grasse[1], Florian Wiesenhofer ◉[1], Niklas Kampleitner[1], Madhumitha Narasimhan ◉[1], Thomas Schmitt ◉[3] & Sylvia Cremer ◉[1] ✉

Treating sick group members is a hallmark of collective disease defence in vertebrates and invertebrates alike. Despite substantial effects on pathogen fitness and epidemiology, it is still largely unknown how pathogens react to the selection pressure imposed by care intervention. Using social insects and pathogenic fungi, we here performed a serial passage experiment in the presence or absence of colony members, which provide social immunity by grooming off infectious spores from exposed individuals. We found specific effects on pathogen diversity, virulence and transmission. Under selection of social immunity, pathogens invested into higher spore production, but spores were less virulent. Notably, they also elicited a lower grooming response in colony members, compared with spores from the individual host selection lines. Chemical spore analysis suggested that the spores from social selection lines escaped the caregivers' detection by containing lower levels of ergosterol, a key fungal membrane component. Experimental application of chemically pure ergosterol indeed induced sanitary grooming, supporting its role as a microbe-associated cue triggering host social immunity against fungal pathogens. By reducing this detection cue, pathogens were able to evade the otherwise very effective collective disease defences of their social hosts.

Medical intervention supports the immune system in fighting disease effectively, yet it can also favour the evolution of new pathogen adaptations, such as resistance to antibiotics[1] or increased virulence[2]. Understanding the selection pressures that facilitate changes in pathogen virulence and transmission trajectories has therefore been a major study focus in evolutionary biology and medicine, revealing that disease ecology and evolution have fundamental implications for health care and epidemiology[1,3–7]. Still, the effects of host defensive behaviours in response to disease, such as pathogen avoidance, self-hygiene or sanitary care by social group members, are far from understood[8–10] despite being ubiquitous and representing strong selective pressures on pathogens by interfering with the course of disease[11–16].

Social insects are excellent models for studying the impact of behavioural care on disease ecology and evolution because the extent of caregiving can be manipulated experimentally[17,18] and the effects monitored over repeated infection cycles. In addition to individual immunity, social insects evolved cooperative disease defences, such as sanitary care and infection treatment, providing 'social immunity' to the colony[19–21]. As one of the first defence lines, nestmates groom pathogen-exposed colony members, thereby removing and disinfecting infectious particles before they can even cause infection[14,15,18,22]. Pathogens that infect their hosts via attaching to and penetrating the body surface, such as many fungal pathogens[23], are therefore under strong selection to counteract the negative impact that mutual grooming of their social hosts has on their fitness. Notably, although grooming uniformly reduces the number of spores that is able to successfully infect the host, the degree of this effect can vary between even closely related fungal species, depending on their specific infection properties[24]. Therefore, social immunity can differentially affect pathogen strains or species when their spores co-infect the same host

[1]ISTA (Institute of Science and Technology Austria), Klosterneuburg, Austria. [2]Laboratory of Evolutionary Genetics, Division of Molecular Biology, Ruđer Bošković Institute, Zagreb, Croatia. [3]Department of Animal Ecology and Tropical Biology, University of Würzburg, Würzburg, Germany. [4]These authors contributed equally: Miriam Stock, Barbara Milutinović. ✉e-mail: barbara.milutinovic@irb.hr; sylvia.cremer@ist.ac.at

individual, so that strains that are outcompeted in individual hosts can in some cases gain competitive advantage in social hosts[24]. By biasing the competitive outcome of pathogen–pathogen interactions, social immunity thus exerts a modulatory effect on co-infecting pathogens. Since co-infections occur very frequently in nature[25,26], social immunity therefore has enormous potential to not only affect the evolution of single pathogens, but also the ecology of pathogen communities.

To experimentally assess the long-term selective effect of social immunity on co-infecting pathogens, we performed a serial passage experiment using six representatives of a natural population of pathogenic *Metarhizium* fungi[24,27] and the invasive Argentine ant, *Linepithema humile*. We subjected the ants to two different selection treatments, one excluding social immunity by keeping exposed ants individually, and the other with social immunity provided by two caregiving nestmates. This allowed for repeated cycles of pathogen–pathogen competition under (1) only individual or (2) individual and social immunity (each in ten replicate lines). For each line, we followed strain success over ten consecutive host infection cycles. We then determined virulence (induced host mortality) and transmission (spore production) characteristics of the selected lines in a common garden experiment, and tested how strongly their spores, as well as the spores' main chemical component ergosterol, elicited grooming in nestmates.

## Results

### Social immunity maintains pathogen diversity longer

We started our serial passage experiment with a balanced mix of six *Metarhizium* strains from two different species, *M. robertsii* (R1–3) and *M. brunneum* (B1–3), all naturally co-existing in a single pathogen population, making co-infections among them probable in the field[24,27]. After exposure, ants were either kept alone (individual selection treatment, $n = 10$ replicate lines) or with caregiving nestmates (social selection treatment, $n = 10$ replicate lines; Fig. 1a). For each infection cycle, we took new ants from our laboratory stock and exposed them to the spores collected from sporulating carcasses of the previous passage of the respective line ($n = 6,312$ exposed ants, 8,026 nestmates), leading to 10 individual and 10 social fungal selection lines.

We observed a loss of the original strain diversity (number of strains per line) in all lines in both selection treatments. From the six strains that were used to start the experiment, only a single prevailed or strongly dominated a second remaining strain after the ten host passages in each line, leading to equally low diversity remaining at the end of the experiment under individual and social immunity (median in both: 1 strain per line; Fig. 1b and Supplementary Table 1). This was despite the fact that social immunity had maintained a higher strain diversity over the course of the selection process, as shown by a significantly higher number of strains still represented per line in passage five (median 1 strain in the individual and 2 strains in the social lines; Fig. 1b and Supplementary Table 1).

We found that under individual host selection, *M. robertsii* considerably outcompeted *M. brunneum* after 5, and completely displaced it after 10 infection cycles. Under social immunity, however, *M. brunneum* was able to persist in 7/10 social lines after 5 infection cycles, and to even fully dominate 2 replicate lines after 10 cycles (Fig. 1b). This can be attributed to *M. brunneum* being less affected by the spore-removing effect of grooming than *M. robertsii*[24], so that the competitive bias repeatedly introduced by the caregiving nestmates allowed it to persist longer in the pathogen community. Nevertheless, *M. robertsii* strain R1 was particularly competitive and was ultimately thriving even under social immunity. The final strain composition under both selection treatments was therefore equally dominated by *M. robertsii* R1, winning in 7/10 individual and 6/10 social lines, followed by a second *M. robertsii* strain (R3; Fig. 1b and Supplementary Table 1). Together, this means that even if both within-line diversity and strain composition within the community did not differ at the end of the experiment, strains in the individual and social treatments had experienced a notably different selection history.

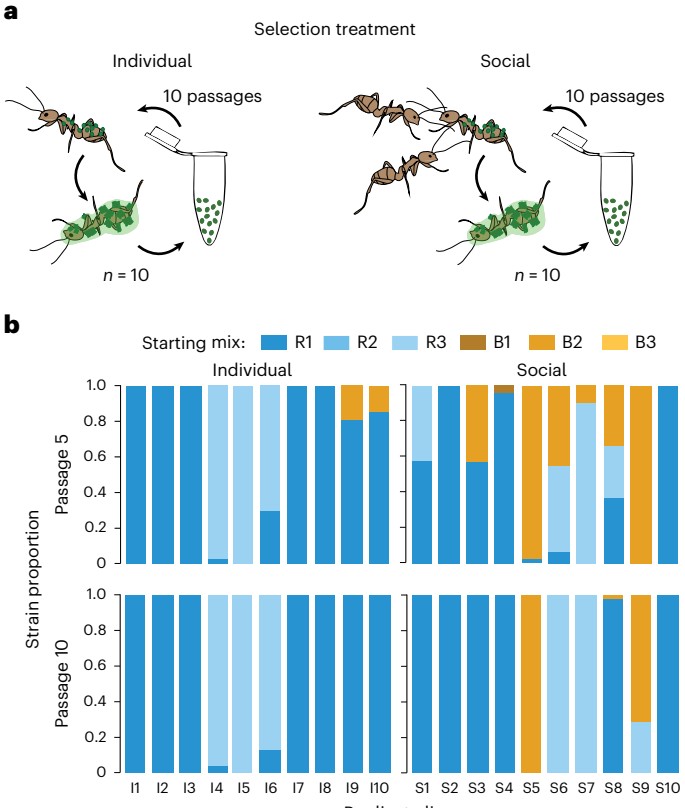

**Fig. 1 | Pathogen selection under individual versus social immunity. a**, Serial passage experiment of fungal pathogens (green) in ants in the absence (left) or presence (right) of caregiving nestmates. Spores collected from carcasses (ants with spore outgrowth) were used for the next infection cycle. **b**, Proportion of the three *M. robertsii* (R1–3) and three *M. brunneum* (B1–3) strains still present from the starting mix after five (top) and ten (bottom) host passages for the individual (I1–10) and social (S1–10) replicate lines. The diversity of strains present per line decreased slower under social immunity (passage 5: WRST, $P = 0.015$; Supplementary Table 1), but reached equally low values of a single remaining or dominating strain per line under both selection treatments at the end of the experiment (passage 10: WRST, $P = 1.000$, Supplementary Table 1). *M. brunneum* was more successful under social immunity. At the end of the experiment, *M. robertsii* strain R1 was dominant in 7 individual and 6 social lines, strain R3 in 3 individual and 2 social lines, and *M. brunneum* strain B2 in 2 social lines, so that the final composition of strains after selection did not differ between the two treatments (Fisher's exact test, $P = 0.629$; Supplementary Table 1). All statistics based on 10 individual and 10 social lines.

The social lines were both constantly exposed to the spore-removing effect of grooming nestmates, and they experienced prolonged pathogen–pathogen competition, particularly in the first infection cycles as an indirect outcome of the presence of social immunity.

### Social immunity lowers virulence and boosts spore production

To test how these distinct selection pressures shaped the pathogens' fitness-relevant parameters, we compared the virulence (induced host mortality) and transmission stage (spore) production between the individual and social lines. To disentangle selection history from the influence of current nestmate presence, we performed a common garden experiment, keeping the ants after exposure under either the matched (individual–individual, social–social) or mismatched (individual–social, social–individual) current host social context ($n = 800$ exposed ants, 800 nestmates).

We found that only the individual lines under current individual ant conditions caused high host mortality (Fig. 2a; Bayesian Multi Level Model, 0.44 mean probability of dying, highest density interval

(HDI) 0.28–0.60, see Supplementary Table 1). This demonstrates that social immunity not only interfered with the expression of virulence in the individual lines under current social host conditions (individual–social), which is a well-described effect of sanitary care[14,15,22], but also caused virulence reduction of the social lines during selection, even when released from this pressure by the absence of nestmates under current individual host condition (social–individual). On the other hand, the continued presence of social immunity selected for high spore production in the social lines (double that of the individual lines), which we found to be independent of current nestmate presence or absence (Fig. 2b and Supplementary Table 1).

Importantly, these results were not confounded by a different number of strains prevailing per line, which would have disparately affected infection outcomes[28], as all lines after selection consisted of only a single or one highly dominant strain (Fig. 1b). Moreover, these results are also significant when considering only the subset of lines in which *M. robertsii* R1 had won the competition (7 individual and 6 social lines, Supplementary Table 2), with the social lines inducing only 65% of the mortality but producing 215% of the spore outgrowth of the individual lines. While statistical testing is only possible for this prominent R1 strain, the 3 individual and 2 social lines in which *M. robertsii* R3 persisted showed similar patterns, with the social lines inducing 75% of the mortality but producing 340% of the spore outgrowth of the individual lines. This suggests that different strains reacted similarly to the selection pressure of social vs individual immunity.

## Pathogens evade social immunity by reduced detection cues

Equal virulence of the social lines under current individual and social host conditions could in principle have resulted either from a better resistance of the spores to removal by grooming, or from caregivers performing less grooming. Our data confirm the latter, as the social lines evoked only 50% of the allogrooming in the nestmates compared with the individual lines (Fig. 3a and Supplementary Table 1; $n = 60$ exposed ants, 120 nestmates, 60 videos of 30 min each). Again, the different *M. robertsii* winner strains showed consistent patterns (reduction to 55% of the grooming elicited by the individual lines in the R1 lines, Supplementary Table 2, and to 35% in the R3 lines). This suggests that the fungi exposed to the selection pressure of social immunity had escaped detection by the caregiving ants.

We therefore analysed whether the spores of the individual and social lines might differ in cues that the ants could use for their detection, as it is known that both pathogen-derived chemicals[29] or the altered surface chemistry of infected or immune-challenged hosts[30–32] can induce disease-preventive behaviours, even if the specific compounds that stimulate sanitary responses are largely unknown[33]. Analysis of the spore chemical profiles using gas chromatography–mass spectrometry revealed 40 spore compounds (Supplementary Table 3). We indeed found 6 of these compounds to be less abundant after selection by social compared with individual immunity (Fig. 3b, Extended Data Fig. 1, and Supplementary Fig. 2 and Table 1); 4 of these represented ergosterol and its derivatives (Supplementary Table 3), which are essential fungal membrane components[34]. Ergosterol was the most prominent compound of the whole spore bouquet, contributing to 60% of the total amount of all spore compounds together, while its derivatives were present only in lower amounts (Fig. 3b and Extended Data Fig. 1). Again, the two successful *M. robertsii* strains showed similar patterns of lowered ergosterol levels in the social lines (to 50% of its abundance in the individual lines in R1, Supplementary Table 2, and to 70% in R3; Fig. 3b). The findings that ergosterol is the main spore compound and showed significantly lower levels after social immunity selection suggest that the reduced allogrooming elicited by the social lines may be mainly linked to ergosterol.

To test whether ergosterol alone—even in the absence of a possible supporting tactile cue from the spores or any other compound of the spore bouquet—would be a sufficient trigger of the ants' sanitary

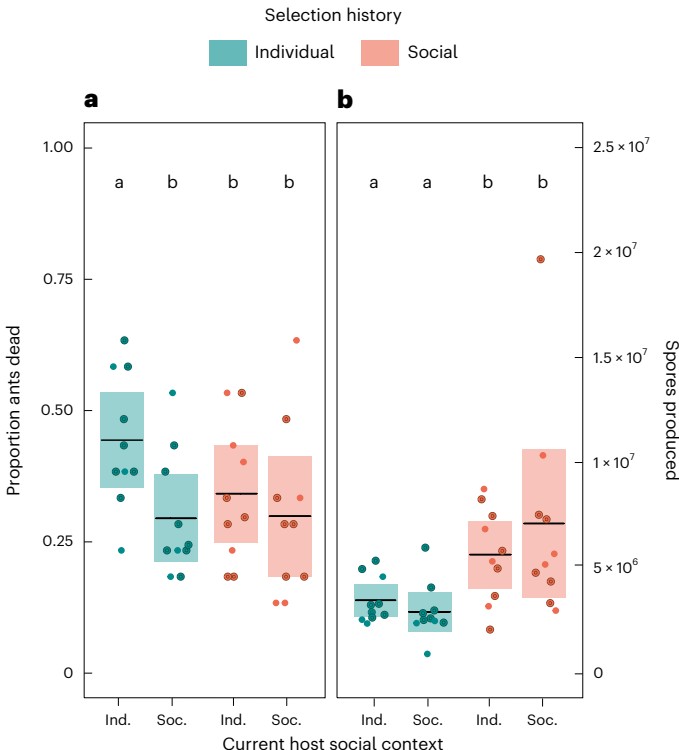

**Fig. 2 | Effect of social immunity on pathogen virulence and transmission.**
**a**, The presence of caregivers in the common garden experiment reduced pathogen virulence, measured as induced host mortality, for the individual lines, while the social lines showed overall low virulence, which was not further reduced by nestmate presence (Bayesian MLM, contrasts Ind-Ind to all other groups < 0.02, see Supplementary Table 1). **b**, Social lines produced about double the amount of spores per carcass than the individual lines, independent of current nestmate presence (LMM, $P < 0.001$, Supplementary Table 1). Dots depict the 20 replicate lines, each tested in two current host conditions (R1-dominant lines indicated by enlarged symbols; for separate statistics see Supplementary Table 2); black lines indicate means and shaded areas indicate 95% CI. Letters denote posthoc differences of $P < 0.05$.

grooming response, we performed a bioassay spraying the pure ergosterol compound onto the ants and quantifying the grooming they received from their nestmates ($n = 22$ ergosterol-treated and 23 sham-treated ants sprayed with acetone only, 45 videos of 30 min each). We found that ergosterol application indeed triggered sanitary grooming (Fig. 3c and Supplementary Table 1), while application of cholesterol ($n = 23$ cholesterol- and 24 sham-treated ants, 47 videos of 30 min each) did not lead to higher levels of allogrooming compared with the control treatment (Fig. 3d and Supplementary Table 1). Together, this revealed that the application of a chemically similar, yet not fungus-derived compound did not show the same response in the ants, and that ergosterol per se is a sufficient chemical cue inducing sanitary caregiving in our experiments.

## Discussion

Our study identified behavioural disease defences as an important selection pressure shaping pathogen communities and their fitness. We started our serial passage experiment with a mix of co-infecting *Metarhizium* strains that were then selected over multiple infection cycles in the presence or absence of host social immunity. We found that social immunity maintained a higher pathogen diversity over the course of the experiment, with the otherwise weaker competitor species *M. brunneum* persisting longer in the presence of nestmate grooming (Fig. 1b). This shows that the modulatory effect of social immunity on infection outcomes in single events of pathogen–pathogen

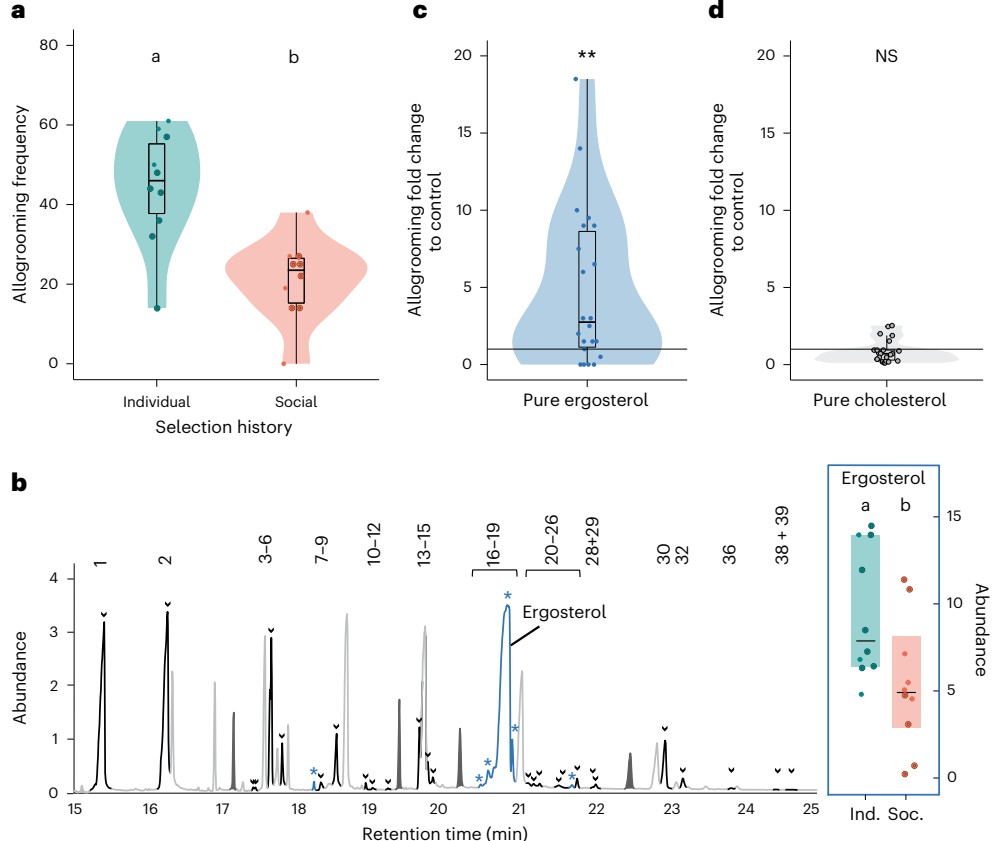

**Fig. 3 | Pathogen evasion of social immunity. a**, Social lines elicited less allogrooming in nestmates (WRST, *P* = 0.003; 20 replicate lines, each tested in 3 biological replicates of 30 min videos each; colours as in Fig. 2). **b**, Their chemical profiles show reduction in six compounds (blue with stars), mostly representing ergosterol (insert; WRST, *P* = 0.025, Supplementary Table 1; dots represent each of the 10 replicate lines, each measured in 3 technical replicates; shaded area: 95% CI around medians (black line)) and its derivatives (Supplementary Fig. 2 and Extended Data Fig. 1; all other fungal compounds black with arrows, compound numbers as in Supplementary Table 3; non-fungal compounds grey, internal standards as filled peaks in dark grey). Abundance in chromatogram given in counts per scan ×10⁷, in ergosterol insert as ISTD response factor. **c**,**d**, Application of pure ergosterol (blue) induced increased nestmate allogrooming over the

sham control (**c**) (WRST, *P* = 0.026, Supplementary Table 1; *n* = 22 ergosterol- and 23 sham-treated replicates, each in 30 min video), whereas this was not the case for the structurally similar, yet not fungus-derived cholesterol (**d**) (grey; WRST, *P* = 0.144, Supplementary Table 1; *n* = 23 cholesterol- and 24 sham-treated replicates, each in 30 min video). Violin- and boxplots in **a**, **c** and **d** show replicate lines as dots, medians as lines within the box defined by the interquartile range, and whiskers revealing minimum to maximum range; kernel density presented as shade in violin plots. Enlarged dots in **a** and **b** indicate R1-dominant lines. Letters denote posthoc differences of *P* < 0.05 between selection histories; \*\**P* < 0.01 vs the sham control; NS, non-significant. Details for all statistical analyses are provided in Supplementary Table 1 (for the subset of R1-dominant lines only, see Supplementary Table 2).

---

competition[24] has the power to shape pathogen communities in the long-term, by inducing substantial changes in pathogen diversity and fitness. Importantly, this could create different eco-evolutionary feedback loops where altered ecological conditions affect host and pathogen evolution in dependence of the host social structure and behaviours deployed[10].

Even if the same strains (*M. robertsii* strains R1 and R3) finally outcompeted their competitors similarly under both selection treatments, our analysis of the pathogen diversity in passage 5 revealed that the strains in the social selection treatment endured prolonged pathogen–pathogen competition over the course of the experiment. In addition to the constant selection pressure of grooming nestmates, this created a second difference in the selection experience for the social vs the individual lines. We found that these distinct selection histories had differently shaped the fitness parameters of the successful pathogen strains at the end of the experiment, namely their virulence and transmission (Fig. 2). Social immunity selection induced increased investment into spore production, yet at the same time reduced pathogen virulence, possibly caused by a resource trade-off[35]. We consider it a plausible scenario that longer pathogen–pathogen competition in the social selection treatment could have sparked investment into

high spore numbers, which is a common outcome of competition in co-infections[24,36]. Higher spore numbers would have proven beneficial in coping with the severe spore-removing effect of grooming[14,15,22], and the continuity of this selection pressure could have amplified the effect over repeated infection cycles.

Interestingly, the spores of the social lines elicited less nestmate allogrooming (Fig. 3a), and they also contained much lower levels of their main spore membrane compound, ergosterol, as well as three of its derivatives (Fig. 3b and Extended Data Fig. 1) than the spores of the individual lines. This suggested a likely role of ergosterol as a fungal detection cue for the ants; a role it could possess across the social insects, as it was also found to act as a grooming enhancer in termites when applied in addition to fungal spores[37]. However, since social insects often respond to changes in the relative compound composition of the overall chemical bouquet rather than to individual compounds[38], these data do not allow pinpointing whether ergosterol alone drives the observed effect, or whether it develops its role only in concert with other compounds, or even with mechanical cues of the spores[39]. By applying ergosterol as a pure chemical in the absence of any other spore cues, we could unequivocally show that ergosterol in itself—but not the structurally similar, animal-derived cholesterol—triggered

sanitary grooming in the ants (Fig. 3c,d). Ergosterol is, therefore, a microbe-associated molecular pattern (MAMP) that induces grooming in ants. Similar to generic disinfection of potentially harmful sites, this comes at a great benefit to the ants in case the detected fungus is indeed a pathogen, yet it comes at very low cost in case it is not. Therefore, reaction to even a broad fungal cue such as ergosterol can help the ants to drastically reduce disease risk. Equally, for a fungal pathogen, reduction of a general fungal compound can be highly beneficial if this increases its chances of successfully infecting the host by not being groomed off.

We therefore show that continued selection under social immunity increased pathogen spore numbers while decreasing their virulence as well as their detection cue levels. We speculate that ergosterol may even be involved as a mechanistic link between transmission, virulence and detection. This is because, as a major spore membrane component, it may become a limited resource when social lines double their spore production, leading to spore membranes with reduced ergosterol levels. Being also important in fungal virulence[34], the low ergosterol levels of the social lines may explain their low virulence; yet given its additional role as a fungal MAMP, this may also allow evasion of the behavioural defences of their hosts, which would otherwise induce high fitness costs to the pathogens.

## Methods
### Ant host
We used workers of the invasive Argentine ant, *Linepithema humile*, as host species. As typical for invasive ants, populations of this species lack territorial structuring and instead consist of interconnected nests forming a single supercolony with constant exchange of individuals between nests[40]. We collected *L. humile* queens, workers and brood in 2011, 2016 and 2022 from its main supercolony in Europe that extends more than 6,000 km along the coasts of Portugal, Spain and France[40–42], from a field population close to Sant Feliu de Guíxols, Spain (41° 49' N, 3° 03' E). Field-collected ants were reared in large stock colonies in the laboratory. For the experiments, we sampled worker ants from outside the brood chambers and placed them into petri dishes with plastered ground (Alabastergips, Boesner, BAG), subjected to their respective treatments as detailed below. Experiments were carried out in a temperature- and humidity-controlled room at 23 °C, 65% relative humidity and a 12 h day/night light cycle. During experiments, ants were provided with ad libitum access to a sucrose-water solution (100 g l$^{-1}$) and plaster was watered every 2–3 d to keep humidity high.

Collection of this unprotected species from the field was in compliance with international regulations, such as the Convention on Biological Diversity and the Nagoya Protocol on Access and Benefit-Sharing (ABS, permit numbers ABSCH-IRCC-ES-260624-1 ESNC126 and SF0171/22). All experimental work followed European and Austrian law and institutional ethical guidelines.

### Fungal pathogens
As pathogen, we used the obligate-killing entomopathogenic fungus *Metarhizium*, whose infectious conidiospores naturally infect ants[43–45] by penetrating their cuticles, killing them and growing out to produce highly infectious sporulating carcasses[23,46]. We used a total of six strains of the two species *M. robertsii* and *M. brunneum*, all isolated from the soil of the same natural population—an agricultural field at the Research Centre Årslev, Denmark[27,47], which makes host co-infections with these sympatric strains in the field likely. As in ref. [24], we used three strains of *M. robertsii* (R1: KVL 12-36, R2: KVL 12-38, R3: KVL 12-35) and three of *M. brunneum* (B1: KVL 13-13, B2: KVL 12-37, B3: KVL 13-14), all obtained from the University of Copenhagen, Denmark (B. M. Steinwender, J. Eilenberg and N. V. Meyling).

We started our selection experiment by exposing the ants to a mix of the six strains in equal proportions. To this end, each strain was grown separately from monospore cultivates from its long-term storage (43% glycerol (Sigma-Aldrich, G2025) in skimmed milk, −80 °C) on SDA plates (Sabouraud-4% dextrose agar, Sigma-Aldrich, 84088-500G) at 23 °C until sporulation. Conidiospores (abbreviated to 'spores') were collected by suspending them in sterile 0.05% Triton X-100 (Sigma-Aldrich, X-100; in milliQ water, autoclaved) and mixed in equal amounts to a total concentration of $1 \times 10^6$ spores ml$^{-1}$. Before mixing, we confirmed that all strains had ≥98% germination.

We exposed worker ants individually to the fungal pathogen by dipping them into the spore suspension using clean forceps (Gebrüder Martin; bioform, B32d). Afterwards, each ant was briefly placed on filter paper (Whatman; VWR, 512-1027) to remove excess liquid before being placed into its experimental Petri dish.

### Serial passage experiment
We tested for the long-term effect of social immunity on pathogen selection, in which the pathogen was serially cycled through the host in the absence or presence of social immunity while the host population remained constant.

**Experimental design and procedure.** After exposure to the fungal spore mix, worker ants were either kept alone (individual host treatment, *n* = 10 replicate lines) or together with two untreated nestmates (social host treatment, *n* = 10 replicate lines; Fig. 1a). Individual ants could only protect themselves by individual immunity (selfgrooming behaviour and their physiological immune system), while the attended ants experienced both individual and social immunity due to the additional allogrooming by their caregiving nestmates. Thus, comparing the two host conditions revealed the effect of social immunity.

As sanitary care by the nestmates reduces the pathogens' success to kill the exposed individuals, we had to set up more experimental dishes of the social host treatment to obtain equal numbers of sporulating carcasses under both selection treatments, from which we then collected the spores for the next host infection cycle. For the individual treatment, we exposed an average of 23 workers per cycle, while an average of 40 workers per cycle were exposed in the social host treatment. The experiment was run for 10 host passages, that is, 27 weeks. In total, 6,312 workers (2,299 in the individual and 4,013 in the social host treatment) were exposed during the course of the experiment, and 8,026 nestmates were used. To obtain the spore suspensions for the next steps, we then collected and pooled the outgrowing spores of the first 8 carcasses produced per replicate line and cycle (that is, a total of *n* = 800 carcasses from the individual and *n* = 800 carcasses from the social host treatment, over the 10 host passages). Dead nestmates were not considered (see below).

In detail, at each host cycle, the freshly exposed ants were placed into Petri dishes with plastered, humidified ground (Ø 3.5 cm for the individual and Ø 6 cm for the social host condition; both Bioswisstec AG, 10035 and 10060) in the absence (individual host treatment) or presence (social host treatment) of two untreated nestmates. We checked survival daily for 8 d. Ants that died within 24 h after exposure were excluded from the experiment as their mortality could not yet have resulted from infection, but rather from treatment procedures. Ants dying from days 2 to 8 were checked for internal *Metarhizium* infections by surface-sterilization (washing the carcass in 70% ethanol (Honeywell; Bartelt, 24194-2.5l; diluted with water) for a few seconds, rinsing it in distilled water, incubating in 3% bleach (Sigma-Aldrich, 1056142500) in sterile 0.05% Triton X-100 for 3 min and rinsing it again three times in water[48]), followed by incubation in a Petri dish on humidified filter paper at 23 °C until day 13, when they were checked for *Metarhizium* spore outgrowth. This timeline was chosen as preliminary work showed that the exposed ants die mostly on days 4 to 8 after exposure (median day 5, for both individual and social host treatments) after exposure and that sporulation required no longer than 5 d in our experimental conditions, so that a duration of 13 d per cycle also allowed for the later dying ants to complete sporulation. Preliminary work further revealed that in cases where nestmates contracted the disease, they died at a delayed

timepoint and with spore outgrowth mostly around the mouthparts. These characteristics were used to distinguish between the directly exposed ants and infected nestmates in the experiment where ants were not colour-marked. The carcasses of sporulating nestmates were excluded from further procedures. An additional control experiment using 120 sham-treated ants showed no *Metarhizium* outgrowth, so that all *Metarhizium* outgrowth in our experiment could be attributed to our experimental infections. Carcasses with saprophytic outgrowth were not considered. For each host passage and each replicate line, we collected the spores of the first 8 ants dying after day 1 from their *Metarhizium*-sporulating carcasses at day 13 in 0.05% Triton X-100, pooled and counted them using an automated cell counter (Cellometer Auto M10, Nexcelom Bioscience). The concentration of each pool was then adjusted to $1 \times 10^6$ spores ml$^{-1}$, and was used directly (that is, in the absence of any intermediate fungal growth step on agar plates) for exposing the ants in the next host infection cycle. The ants of each host passage were thus dipped in the same spore concentration. The remaining spore suspension was frozen at −80 °C in a long-term storage for further analysis.

**Pathogen diversity and strain composition.** We analysed which strains were present and in which proportion after 5 and 10 passages in each of the 10 individual and 10 social replicate lines. To this end, we first extracted total DNA from the respective spore pools ($n = 40$), which we analysed (1) quantitatively for the respective representation of *M. robertsii* vs *M. brunneum* (using species-specific real-time PCR targeting the PR1-gene sequence; detailed below) and (2) qualitatively for which of the 6 original strains were still present in the pool (using strain-specific microsatellite analysis; detailed below). We used this first estimate of remaining strain diversity and composition of each pool to determine how many spores we had to analyse separately for their strain identity after individualization by FACS sorting and growing them individually as colony forming units (c.f.u.s). This clone-level strain identification was again performed using microsatellite analysis ($n = 1,347$ individualized clones from the 40 spore mixes, in addition to $n = 27$ spores from the 6 ancestral strains; detailed below). Such clonal separation was needed since expansion of the spore mix by growth on SDA plates was not representative of the genetic composition of the strains in the pool, due to strong strain–strain growth inhibition when growing in a mix.

In detail, we extracted the DNA of the 6 ancestral strains and the 40 spore mixes (10 each for individual and social lines at passages 5 and 10), as well as of 27 individualized clones of the ancestral strains and 1,374 clones from the 40 pools of passages 5 and 10, by centrifuging 100 µl of the spore suspensions in 1.5 ml tubes (Eppendorf, 0030120086) at full speed for 1 min and discarding the supernatant. Nuclease-free water (50 µl) was added and the spores were crushed in a bead mill (Qiagen TissueLyser II, 85300) at 30 Hz for 10 min using acid-washed glass beads (425–600 µm; Sigma-Aldrich, G8772). DNA was extracted using a DNeasy blood and tissue kit (Qiagen, 69506) following the manufacturer's instructions, using a final elution volume of 50 µl buffer AE.

For the quantitative species-level analysis of the pools, we performed quantitative real-time PCR (qPCR) using primers and differently labelled probes[24] that we had developed on the basis of the sequence of the PR1 gene[49] (forward: 5′ TCGATATTTTCGCTCCTG, reverse 5′-TTGTTAGAGCTGGTTCTGAAG, PR1 probe *M. brunneum*: 5′-(6-carboxyfluorescein (6FAM))TATTGTACCTACCTCGATAAGCTTAG AGAC(BHQ1), PR1 probe *M. robertsii*: 5′-(hexachloro-fluorescein (HEX)) AGTATTGTACCTCGATAAGCTCGGAGAC(BHQ1)). Reactions were performed in 20 µl volumes using 10 µl iQ Multiplex Powermix (Bio-Rad, 1725849), with 600 nM of each primer (Sigma-Aldrich), 200 nM of each probe (Sigma-Aldrich) and 2 µl of extracted DNA. The amplification programme was initiated with a first step at 95 °C for 3 min, followed by 40 cycles of 10 s at 95 °C and 45 s at 60 °C. Primer efficiency was above 92% for both primer/probe combinations using standard curves of

10-fold dilutions of known input amounts. Data were analysed using Bio-Rad CFX Manager software.

For the strain-specific analysis of both the pools and the individualized clones, we used two microsatellite loci, *Ma307*[50] and *Ma2054*[51]. Microsatellite locus *Ma307* (forward: 5′-(6FAM)CATG CTCCGCCTTATTCCTC-3′, reverse: 5′-GGGTGGCGAAGAAGTAGACG-3′) allowed distinction of all strains except two of the *M. brunneum* strains (B1 and B3), which were distinguished by microsatellite locus *Ma2054* (forward: 5′-(6FAM)GCCTGATCCAGACTCCCTCAGT-3′, reverse: 5′-GC TTTCGTACCGAGGGCG-3′). We analysed the microsatellites by E-Gel high-resolution 4% agarose gels (ILife Technologies, G501804) and fragment length analysis (done by Eurofins MWG) using Peak Scanner software 2.

For clone individualization, we used flow cytometry to sort single spores out of the 40 spore pools (and the 6 ancestral strains for comparison) on 96-well plates (TPP; Biomedica, TP-92696) containing SDA (100 µl per well). The unstained spore population was detected using the FSC (forward scatter)/SSC (side scatter) in linear mode (70 µm nozzle, FACS ARIA III, BD Biosciences, as exemplified in Supplementary Fig. 1). Purity mode was set to 'single cell' and spore clones were obtained by sorting 1 particle event into each well. Sorting and data analysis were performed using Diva 6.2 software. The number of spores that we obtained for microsatellite analysis varied for each replicate, as it was adjusted to the remaining strain diversity estimate that we obtained from the quantitative and qualitative analysis of the pools. In total, we analysed 4–5 clones per ancestral strain (total $n = 27$) and a median of 5, but up to 101 different clones for the pools (total $n = 1,347$), as we intensified analysis for the strains that were revealed to be present at low frequency on the basis of previous analysis.

## Common garden experiment

**Experimental design and procedure.** We then tested whether the successful lines at the end of the experiment (that is, after 10 host passages) differed in their virulence (induced host mortality) and investment into transmission stages (produced spore number) depending on their selection history (individual vs social), when current host social context either reflected the selection history or not. This common garden experiment thus led to 20 matched combinations of selection history and current condition (10 each of the individual lines in current individual host conditions (individual–individual) and the social lines in current social host conditions (social–social)) and 20 non-matched conditions (10 each of the individual lines in current social host conditions (individual–social) and the social lines in current individual host conditions (social–individual)).

We obtained the lines for performance of the common garden experiment by the following procedure: (1) for the 16 out of the 20 replicate lines, where a single strain was the sole remaining representative at the end of the experiment (Fig. 1b), we expanded one of the c.f.u.s grown after FACS sorting (see above) by plating on SDA; (2) for the 4 remaining replicates in which two strains had remained (two individual and two social replicate lines), we expanded one c.f.u. of each of the remaining strains and mixed the spores in their representative proportion, as determined above.

**Virulence and transmission.** For the 10 individual and 10 social lines, we determined the induced host mortality as a measure of virulence and the outgrowing spore number as transmission stage production under their matched and non-matched current host conditions. We exposed the workers as in the selection treatment, kept them either alone or with two untreated nestmates, and monitored their mortality daily for 8 d. Again, ants dying in the first 24 h after treatment and dying nestmates were excluded from the analysis. In total, we obtained survival data of 797 ants (19–20 ants exposed for each of the 10 replicates from each of 4 combinations of selection history and current host condition). Dead ants were treated as above and their outgrowing spores collected by a

needle dipped in sterile 0.05% Triton X-100 directly from the carcass, and resuspended in 100 µl of sterile 0.05% Triton X-100. The number of spores per carcass was counted individually using the automated cell counter, as described above (n = 215; median of 5 per replicate). We excluded one outlier carcass (from replicate I5) where we expected a counting error as this single carcass showed approx. 100-fold higher spore count than the other carcasses of this replicate. Exclusion of this outlier did not affect the statistical outcome. The proportion of ants dying per replicate line for each combination of selection history and current host condition and the number of spores produced by all carcasses per replicate were respectively used as measures of virulence and transmission (mean carcass spore load per replicate plotted in Fig. 2).

### Allogrooming elicitation by the fungal lines

We determined the allogrooming elicited by the individual and the social lines. To this end, we exposed workers as above and observed the allogrooming performed by two untreated nestmates towards the exposed ant. In detail, we performed 3 biological replicates for each of the 20 replicate lines (n = 10 individual and 10 social lines, resulting in a total of 60 videos), where the exposed ant was placed with two untreated nestmates within 10 min after exposure, and filmed with Ueye cameras for 30 min (whereby 4 cameras were used in parallel, each filming 3 replicates simultaneously, and using StreamPix 5 software (NorPix 2009-2001) for analysis). Videos were obtained in a randomized manner and labels did not contain treatment information so that the observer was blind to both the selection history and individual treatment during the behavioural annotations. For each ant, we observed both self- and allogrooming. Start and end times for each grooming event were determined, supported by use of the software BioLogic (Dimitri Missoh, 2010 (https://sourceforge.net/projects/biologic/)).

As the ants in our serial passage and common garden experiments were not colour-marked, we also used unmarked ants for this behavioural experiment to keep conditions the same. This was possible as preliminary data with colour-coded nestmates (n = 18 videos) had shown that exposure alters the ant's behaviour and that of its untreated nestmates in a predictable way that allows reliable classification of the pathogen-exposed individuals from the untreated nestmates; we used the following rules to classify an ant as the exposed individual: (1) the individual spent >5% more time (of the 30 min observation period) selfgrooming than the other individuals; (2) if the difference in selfgrooming time between the individuals was <5%, the ant receiving the highest duration of allogrooming was classified as the exposed individual. These rules showed 89% correct assignments in the test videos. Hence, given that we repeated each fungal line in 3 biological replicates, the probability of misclassifications in all 3 replicates per line was only 0.1%. The numbers of allogrooming events of the 3 replicates per line were summed and used to compare allogrooming elicitation after the 10 host passages.

### Chemical analysis of the fungal lines

We used gas chromatography–mass spectrometry (GC–MS) to determine the chemical compound composition of the 24 clones obtained from the lines resulting from the serial passage experiment (that is, 1 clone each from the 16 pure lines and 2 clones each from the 4 lines with 2 remaining strains; each run in 3 technical replicates, as detailed below).

To this end, we collected the clones after growth on SDA plates in sterile 0.05% Triton X-100. For each clone, a 200 µl spore suspension ($1 \times 10^9$ spores ml$^{-1}$) was centrifuged to obtain a spore pellet, which was then washed with water three times. The resuspended spores were dried under a gentle nitrogen stream for 2–2.5 h in brown glass vials (Markus Bruckner, BA10513). From this, spore compounds were extracted in 350 µl n-pentane (SupraSolv, Supelco; VWR, 1.07288.2500) under gentle agitation at room temperature for 5 min and the extracts

were then centrifuged at 3,000 g for 5 min. Each sample was then split into 3 technical replicates to be run separately on the GC–MS system by transferring 80 µl of the supernatant into 3 separate glass vials (Markus Bruckner, BA10230), which were then sealed with aluminum-faced silicon septa (Markus Bruckner, BA10176). In addition, handling controls containing only n-pentane were prepared simultaneously and treated alongside the biological samples.

The solvent contained 8 internal standards (n-octane, n-decane, n-dodecane, n-octadecane, n-tetracosane, n-triacontane, n-dotriacontane and n-hexatriacontane, all fully deuterated; CDN Isotopes, D-0451, D-0960, D-0882, D-0956, D-0883, D-2708, D-0973 and D-0950) at 0.5 µg ml$^{-1}$. The extracts were run in a randomized manner, intermingled with blank runs (containing only n-pentane) and handling controls, using GC–MS (GC7890 coupled to MS5975C; Agilent Technologies). A liner with one restriction ring filled with borosilicate wool (Joint Analytical Systems) was installed in the programmed temperature vaporization (PTV) injection port of the GC, which was pre-cooled to −20 °C and set to solvent vent mode. Of the sample extractions, 50 µl were injected automatically into the PTV port at 40 ml s$^{-1}$ using an autosampler (CTC Analytics, PAL COMBI-xt, CHRONOS 4.2 software; Axel Semrau) equipped with a 100 µl syringe. Directly after injection, the temperature of the PTV port was increased to 300 °C at 450 °C min$^{-1}$, thereby vaporizing the sample analytes and transferring them to the column (DB-5ms; 30 m × 0.25 mm, 0.25 µm film thickness; 122-5532UI) at a flow rate of 1 ml min$^{-1}$.

To ensure optimal peak separation and shape, helium was used as the carrier gas at a constant flow rate of 3.0 ml min$^{-1}$ for 2 min, then ramped down to 1.1 ml min$^{-1}$. The oven was programmed to hold 35 °C for 4.5 min, then ramped to 325 °C at 20 °C min$^{-1}$ and held this temperature for 10 min. The GC–MS transfer line was set to 325 °C, and the mass spectrometer operated in electron ionization mode (70 eV) with an ion source temperature of 230 °C and a quadrupole temperature of 150 °C, with a detection threshold of 150 and mass scan range of 35–600 amu. The data were acquired using MassHunter Workstation, Data Acquisition software B.07.01 (Agilent).

In addition, a $C_7$-$C_{40}$ saturated alkane mixture (0.1 µg ml$^{-1}$ for each alkane in pentane; Sigma-Aldrich, 49452-U) was run, enabling calculation of Kováts retention indices (RIs) and correcting for eventual shifts in retention time during the time the samples were run. Compound peaks were extracted from the total ion current chromatograms (TICCs) of representative sample runs, using a deconvolution algorithm (MassHunter Workstation, Qualitative Analysis B.07.00; Agilent). Most abundant characteristic ions of the detected spore compounds were used as quantifier ions to establish a quantification method, enabling automated integration of the peak area of each compound, and calculating its relative amount compared to the internal standard eluting just before (MassHunter Workstation, Quantitative Analysis B.07.00; Agilent) to correct for eventual injection differences, thereby obtaining the internal standard (ISTD) response factor. Note that for each sample, only compounds with a signal to noise ratio equal to or higher than 10 were considered.

Compound annotation was performed by comparing the mass spectra and Kováts RI of each compound to the Wiley 9th edition/NIST 11 combined mass spectral database (National Institute of Standards and Technologies) and by manual interpretation of diagnostic ions. For each compound, we provide the confidence level of compound annotations, following the Compound Identification work group of the Metabolomics Society[52]. Compound annotation (confidence level 1) for ergosterol was performed after injecting a standard solution containing pure ergosterol (Acros Organics, ≥98% purity; VWR, 49452-U) in different concentrations and comparing the RI and mass spectrum to the peaks in the biological samples. For the unknown compound 7 (RI 2,631), we show the mass spectrum in Supplementary Fig. 3.

The 40 obtained compounds and their classifications are given in Supplementary Table 3. Figure 3b depicts a typical chromatogram

showing compound abundance (×10⁷ counts). Note that not all compounds were found to be detectable at or above the signal to noise ratio of 10 in each sample. In Fig. 3b, we display the abundance of ergosterol standardized to its closest internal standard as ISTD response factor, by the mean of the 3 replicate measures per fungal line. We also calculated the relative abundance of ergosterol, the most prominent compound of the whole bouquet, as the proportion of its abundance to the total abundance of all 40 spore compounds per line (mean of the 3 replicates).

### Allogrooming elicitation by pure ergosterol

To test whether ergosterol in its pure form (in the absence of any accompanying compound of the spore bouquet and in the absence of any spore-derived tactile cues) could trigger allogrooming by nestmates, we performed a bioassay in which we applied pure ergosterol dissolved in acetone (as detailed below) to ant workers and quantified the allogrooming that ergosterol-treated ants received in comparison to sham-treated controls (treated with pure acetone). To further determine the specificity of the ants' response to ergosterol, we tested for a possible allogrooming-inducing effect of a structurally similar, yet not fungus-derived compound, pure cholesterol (see below).

**Ergosterol application onto the ants.** We dissolved ergosterol (≥98% purity; VWR, ACRO117810250) in acetone (Sigma-Aldrich, 65051) at a concentration of 0.5 mg ml⁻¹ in an ultrasonic bath for 20 min in the dark. The ergosterol solution was then aliquoted into brown 10 ml perfume spray bottles equipped with a 360° rotation nozzle. Single ant workers were treated by application of one spray stroke of the ergosterol solution or a pure acetone control from a standardized height. The treated ant was put onto filter paper until it was dry and had recovered from treatment, showing normal running behaviour.

**Quantification of the ergosterol application.** To determine how many spore equivalents the sprayed-on ergosterol solution was equal to, we determined the ergosterol level for (1) ants that were sprayed with the ergosterol solution used in the bioassay, (2) ants exposed to on average of 10³ spores of *M. robertsii* R1 and (3) control-treated ants without any spores. Using the methods described below, we quantified for each treatment both the ergosterol abundance (absolute abundance in relation to the internal standard as detailed below, that is the ISTD response factor) per ant by the highly sensitive selective ion monitoring (SIM) method in the GC–MS, and the absolute spore number per exposed ant by droplet digital PCR (ddPCR). This revealed an average ISTD response factor of 0.00396 per ant exposed to 10³ spores, and a baseline value of 0.00081 for the sham-treated ants, probably reflecting natural occurrence of fungal saprophytes in the nest boxes (note that these baseline values are only detectable with the highly sensitive SIM method, and only in pools of 8 ants, as they are below detection threshold when scan mode was used, as well as in SIM-analysed individual ants). Deducting this baseline from the ergosterol abundance in the spore-exposed ants revealed an ergosterol ISTD response factor per 1,000 spores of 0.00315. Comparing this to the value of 0.95310 (the ISTD response factor of the ergosterol-sprayed ants minus the baseline) revealed that we applied the ergosterol proportion (that is, 60% of the total compound abundance per spore) of an equivalent of $3 \times 10^5$ spores. Therefore, our bioassay represented a realistic infection dose for ants, for example when in contact with sporulating carcasses of their own species or their insect prey, since carcassess sporulating with *Metarhizium* are covered with several to even hundreds of millions ($10^6$–$10^9$) of spores per carcass, packed into dense spore packages, making bulk transfer of high spore numbers likely[53–55].

**Qualitative and quantitative ergosterol determination via sensitive SIM GC–MS.** Directly after treatment, the ants were frozen in pools of 8 (in 3 replicates for the ergosterol spray and natural baseline,

6 replicates for the spore-exposed ants) and kept at −80 °C for later chemical analysis. To this end, each pool was individually extracted with 110 µl pentane (Merck; VWR, 1.07288.1000) containing an internal standard (1 µg ml⁻¹ C24d50 in pentane to monitor variations from sample run to sample run) under gentle agitation at room temperature for 5 min. Of each extract, 90 µl was transferred into a glass vial with integrated glass inserts (Klaus Trott, 40 11 00 768), closed by polytetrafluoroethylene-faced silicon septa (Klaus Trott, 3111Y1015). We also prepared solvent blanks containing only pentane.

The extracts gained were analysed using the same GC–MS system and methods as described above for the spore suspensions, yet here we chose the SIM mode to also detect small amounts of ergosterol ($m/z$ 363.3 was selected as quantifier and $m/z$ 396.3 as qualifier ion). For detection of the internal standard, SIM was also applied for the ion with $m/z$ 66.2. An ergosterol standard solution (2.5 µg ml⁻¹ in pentane, analysed in full scan and SIM modes) was used to identify ergosterol in the samples by comparing its RI and the qualifier and quantifier ions. The ion with $m/z$ 363.3 was used as quantifier to establish a quantification method, enabling automated integration of the peak area of ergosterol, and calculating its ISTD response factor, that is, its amount relative to the internal standard to correct for injection differences.

**Absolute spore quantification by ddPCR.** To be able to refer the absolute ergosterol levels of the spore-exposed ants to their absolute spore numbers, we established a sensitive ddPCR method on the basis of the single-copy PR1 gene[56], so that copy number that is determined by ddPCR as an absolute value directly equals spore number. The DNA extraction and ddPCR protocol for this direct quantification of spores on ants deviated from the above-described real-time PCR performed on pure spore suspensions in that the samples ($n = 16$ pools of 5 ants each) were homogenized using a TissueLyser II (Qiagen, 85300) with a mixture of one 2.8 mm ceramic (VWR, MOBO13114-325), five 1 mm zirconia beads (BioSpec Products; Lactan, N0381) and approx. 100 mg of 425–600 µm glass beads (Sigma-Aldrich, G8772) in 50 µl water. Total DNA was extracted using the DNeasy blood and tissue kit (Qiagen, 69506) according to the manufacturer´s recommendations, with a final elution volume of 50 µl buffer AE. The genomic DNA was digested using EcoRI-HF and HindIII-HF enzymes (both New England Biolabs, R3101S and R3104S) within the 20 µl 1x ddPCR reaction, comprising: 10 µl 2× ddPCR Supermix for probes (Bio-Rad, 1863010), 18 pmol of both PR1 primers (forward primer and reverse primer as given above for the qPCR; Sigma-Aldrich), 5 pmol of the *M. robertsii* PR1 probe (as above, yet labelled with 6FAM), 10 U each of EcoRI-HF and HindIII-HF, 4.8 µl nuclease-free water (Sigma-Aldrich, W4502-IL) and 2 µl DNA template. Droplet generation was done using the QX200 droplet generator (Bio-Rad, 17005227) according to the manufacturer's instructions.

Droplets were transferred into a 96-well plate (Eppendorf, 0030128575) for PCR amplification in a T100 thermal cycler (Bio-Rad, 1861096). Cycling conditions were as follows: enzyme activation for 10 min at 95 °C, followed by 40 cycles of 30 s at 94 °C and 1 min at 56 °C, then enzyme deactivation for 10 min at 98 °C. For the entire protocol, the ramp rate was set to 2 °C s⁻¹. Following PCR amplification, the PCR plate was put into a QX200 droplet reader (Bio-Rad, 1864003) for the readout of positive and negative droplets. Data analysis was done using the QuantaSoft Analysis Pro software (Bio-Rad). The thresholds were set manually to 3,500. The absolute number of spores per sample was computed from the obtained copy number per well by adjusting to the amount of template used in the ddPCR reaction (2 µl), the elution volume used during DNA extraction (50 µl) and the dilution factor in the ddPCR.

**Quantification of allogrooming behaviour.** To test how much allogrooming would be elicited by ergosterol vs sham application, we observed the behaviour of two individually colour-coded nestmates (by application of a dot on the gaster with marker pens (uniPOSCA,

POSCA) on the day before the experiment) towards individual ants that had been treated either with acetone only ($n = 23$) or with the ergosterol solution ($n = 22$; ants treated with the chemicals were not colour-marked). Immediately after treatment, the ergosterol- or sham-treated individual was added to two nestmates in a round plastic box (Ø 3.5 cm; Bioswisstec, 10035) with humidified plastered floor and Fluon (Whitford, GP1-CLPBB1K)-coated walls, using forceps cleaned with methanol, followed by acetone. Videos were recorded for 30 min each (2 replicates per camera, 4 cameras in parallel, using digital microscope cameras (MicroDirect 1080p HD, Celestron) with VirtualDub software v1.10.4 (http://www.virtualdub.org/)). Videos were obtained in a randomized manner and labels did not contain treatment information so that the observer was blind to the application of the treated, unlabelled individual during behavioural observations. The number of allogrooming events towards the treated individual was noted per replicate as above, except here using manual annotation.

**Specificity of the allogrooming induction by ergosterol.** To evaluate whether cholesterol, as a non-fungal-derived compound with high structural similarity and similar molecular mass to the fungus-specific ergosterol, would similarly elicit increased allogrooming over the baseline levels of the acetone sham control, we repeated the bioassay exactly as above, except for the use of cholesterol (≥99% purity; Sigma-Aldrich, C8667-1G) instead of ergosterol. We dissolved the cholesterol in acetone to reach the same concentration (0.5 mg ml$^{-1}$), sprayed it on single ants with the same procedure, and determined the allogrooming of two nestmate ants on the cholesterol-treated ant ($n = 23$) or the acetone-treated sham controls ($n = 24$) in 30 min as above (4 replicates per camera).

**Statistical data analysis**
Data generated in this study are provided as source data. Statistical analyses were performed using R v4.0.5[57]. Residual diagnostics of regression models (qq-plots for deviations from the expected distribution, tests for overdispersion, outliers, plots of model residuals against the predicted values) were tested using the DHARMa package[58]. To obtain normality (for spore production in Fig. 2b, untransformed values are shown), we used the package bestNormalize[59] to transform data using arcsinh transformation. Effect sizes were calculated using the packages effectsize[60] and rstatix[61].

**Pathogen diversity and strain composition.** We determined whether strain diversity (that is, the number of strains prevailing per line) differed between the two selection treatments, both during (passage 5) and at the end of the experiment (passage 10), using non-parametric Wilcoxon Rank Sum tests (WRST) for independent samples. For the strains persisting at the end of the experiment (that is, strains R1, R3 and B2), we analysed whether their representation (presence or absence) differed between the individual and social lines using a Fisher exact test based on a 3 × 2 contingency table to determine whether final strain composition of the lines successful in the serial passage experiment differed between the individual and social immunity treatment.

**Pathogen virulence and transmission.** For all lines after serial passaging, we analysed virulence (proportion of ants dead) and transmission potential (number of spores produced) in the common garden experiment. For both, we first tested whether there existed a significant interaction of our two main effects: 'selection history × current host social context'. To this end, we fitted a generalized linear mixed model (GLMM) with binomial error terms and logit-link for the proportion of dead ants (virulence) and a linear mixed model (LMM) for spore production, as the spore numbers were normally distributed after transformation (Fig. 2, using the lmer and glmer functions in the lme4 package[62]). Both models were fitted including 'replicate

line' and 'laboratory stock colony' as random effects. For the virulence data, the GLMM model showed singular fit, indicating that the model was overfitted. We therefore applied Bayesian modelling in Stan computational framework (http://mc-stan.org/) using the brms package[63,64]. Posteriors were sampled from six Hamilton Monte Carlo (HMC) chains, 5,000 iterations and a warmup of 2,500 iterations, with target acceptance rate (adapt_delta) of 0.99 and default priors. We verified that all six chains converged by visually checking the trace plots. For comparisons between parameters, we estimated the difference between the posterior samples and reported the mean differences, CI and the level of support of the difference in the parameters. For the spore production model (LMM), we found that both the interaction and the full model compared to a null model (including the intercept only) were significant, using likelihood ratio tests[65]. Therefore, we obtained all-pairwise posthoc comparisons by fitting the same model with four groups combining selection history and the current host social context (individual-individual, social–social, individual–social, social–individual), and corrected the P values for multiple testing using the Benjamini-Hochberg[66] correction to protect against a false discovery rate (FDR) of 5%. Adjusted two-sided P values are reported (Supplementary Table 1). Same statistics were performed for the subset of the R1-dominant lines only (highlighted symbols in Fig. 2; statistical results in Supplementary Table 2). Note that for the spore numbers of the R1-dominant lines, the interaction was not significant, hence we also show the significance of the main effects.

**Allogrooming.** We tested for differences in the number of allogrooming events elicited in nestmates (1) between the individual vs social lines (Fig. 3a) and (2) between the sham-treated vs ergosterol- (Fig. 3c) or cholesterol- (Fig. 3d) treated ants in the two bioassays, using non-parametric WRST for independent samples. Note that in Fig. 3c,d we performed the statistics on the raw data but display the fold change of allogrooming for each treatment group relative to its acetone control. For each of the individual and social lines (Fig. 3a), we obtained a single sum of allogrooming events from the three biological replicates. Statistics are reported in Supplementary Table 1; for allogrooming elicitation of the subset of the R1-dominant lines only (highlighted symbols in Fig. 3a), see Supplementary Table 2.

**Chemical data analysis.** To analyse the differences in spore chemical composition of individual and social lines after passage 10, we first performed a permutational multivariate analysis of variance (PERMANOVA; Supplementary Table 1) using the adonis function in the vegan package and a Euclidean distance matrix[67]. To identify the individual compounds that differed between the two selection treatments, we performed a conditional random forest classification ($n$ trees = 500, $n$ variables per split = 6) using the randomForest package[68]. For the selection of important compounds, we used the absolute value of the lowest negative score of the mean decrease in accuracy as a threshold (Supplementary Fig. 2). The six compounds that were identified as important contributors to treatment differences were subsequently analysed with separate WRST for independent samples, and the P values corrected for multiple testing with the Benjamini-Hochberg method, as detailed above (Fig. 3b and Extended Data Fig. 1). Statistics are reported in Supplementary Table 1; for the subset of the R1-dominant lines only (highlighted symbols in Fig. 3b and Extended Data Fig. 1), see Supplementary Table 2.

**Reporting summary**
Further information on research design is available in the Nature Portfolio Reporting Summary linked to this article.

## Data availability
All data generated in this study are provided with this paper as source data in a single Excel file (Stock_Milutinovic_source_data_xlsx).

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

## Acknowledgements

We thank B. M. Steinwender, N. V. Meyling and J. Eilenberg for the fungal strains; J. Anaya-Rojas for statistical advice; the Social Immunity team at ISTA for ant collection and experimental help, in particular H. Leitner, and the ISTA Lab Support Facility for general laboratory support; D. Ebert, H. Schulenburg and J. Heinze for continued project discussion; and M. Sixt, R. Roemhild and the Social Immunity team for comments on the manuscript. The study was funded by the German Research Foundation (CR118/3-1) within the Framework of the Priority Program SPP 1399, and the European Research Council (ERC) under the European Union's Horizon 2020 Research and Innovation Programme (No. 771402; EPIDEMICSonCHIP), both to S.C.

## Author contributions

S.C., M.S., B.M., M.H. and T.S. contributed to the conception of the project. Data were generated by M.S., A.V.G., F.W., M.N. and M.H., and analysed by B.M., M.S., N.K., M.H., A.V.G. and T.S. The manuscript was written by B.M. and S.C. with support from M.H., A.V.G. and T.S. and approved by all authors. Funding was obtained by S.C.

## Competing interests

The authors declare no competing interests.

## Additional information

**Extended data** is available for this paper at https://doi.org/10.1038/s41559-023-01981-6.

**Correspondence and requests for materials** should be addressed to Barbara Milutinović or Sylvia Cremer.

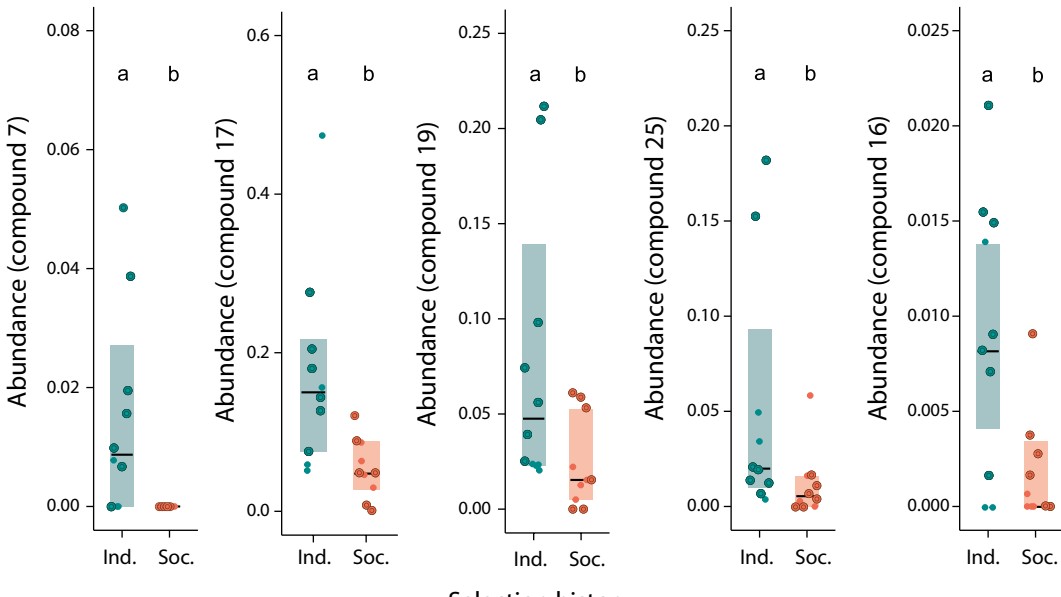

**Extended Data Fig. 1 | Abundance of the five additional spore compounds affected by social immunity.** Further to ergosterol (compound 18; Fig. 3b insert), five additional compounds were identified by the random forest as important in differentiating between the two selection histories (shown in the order of their derived compound importance; Supplementary Fig. 2). Three of these (compounds 16, 17, 19) were classified as ergosterol derivatives, one as an ester (compound 25) and one is unknown (compound 7; for mass spectrum see Supplementary Fig. 3). All of the identified important compounds were significantly lower in abundance in the social than the individual lines (WRST, comp. 7 and 17, p = 0.007; comp. 19, p = 0.025; comp. 25 and 16: p = 0.031; Supplementary Table 1; significant differences denoted by different letters). Dots represent the 10 replicate lines each of the individual and social selection history (with enlarged dots indicating R1-dominant lines; for separate statistics see Supplementary Table 2), shades the 95% CIs around the medians (black lines). Colours as in Fig. 2.

# Reporting Summary

## Statistics

For all statistical analyses, confirm that the following items are present in the figure legend, table legend, main text, or Methods section.

| n/a | Confirmed | |
|---|---|---|
| ☐ | ☒ | The exact sample size (*n*) for each experimental group/condition, given as a discrete number and unit of measurement |
| ☐ | ☒ | A statement on whether measurements were taken from distinct samples or whether the same sample was measured repeatedly |
| ☐ | ☒ | The statistical test(s) used AND whether they are one- or two-sided *Only common tests should be described solely by name; describe more complex techniques in the Methods section.* |
| ☒ | ☐ | A description of all covariates tested |
| ☐ | ☒ | A description of any assumptions or corrections, such as tests of normality and adjustment for multiple comparisons |
| ☐ | ☒ | A full description of the statistical parameters including central tendency (e.g. means) or other basic estimates (e.g. regression coefficient) AND variation (e.g. standard deviation) or associated estimates of uncertainty (e.g. confidence intervals) |
| ☐ | ☒ | For null hypothesis testing, the test statistic (e.g. $F$, $t$, $r$) with confidence intervals, effect sizes, degrees of freedom and $P$ value noted *Give P values as exact values whenever suitable.* |
| ☐ | ☒ | For Bayesian analysis, information on the choice of priors and Markov chain Monte Carlo settings |
| ☐ | ☒ | For hierarchical and complex designs, identification of the appropriate level for tests and full reporting of outcomes |
| ☐ | ☒ | Estimates of effect sizes (e.g. Cohen's *d*, Pearson's *r*), indicating how they were calculated |

*Our web collection on statistics for biologists contains articles on many of the points above.*

## Software and code

Policy information about availability of computer code

| | |
|---|---|
| Data collection | software used for real-time PCR: Bio-Rad CFX Manager software, for droplet digital PCR: QuantaSoft™ Analysis Pro Software (Bio-Rad); for microsatellite analysis: Peak Scanner Software 2; for FACS sorting: Diva 6.2 software; for videos: StreamPix 5 software and VirtualDub software v 1.10.4; for behavioural annotation: BioLogic; for gas chromatography-mass spectrometry: CHRONOS 4.2 software, Axel Semrau and MassHunter Workstation, Data Acquisition software B.07.01 and Qualitative and Quantitative Analysis B.07.00; Agilent Technologies; |
| Data analysis | statistics performed with R version 4.0.5, packages: DHARMa, bestNormalize, lme4, brms, vegan, randomForest, effectsize, rstatix |

For manuscripts utilizing custom algorithms or software that are central to the research but not yet described in published literature, software must be made available to editors and reviewers. We strongly encourage code deposition in a community repository (e.g. GitHub). See the Nature Portfolio guidelines for submitting code & software for further information.

## Data

Policy information about availability of data

All manuscripts must include a data availability statement. This statement should provide the following information, where applicable:
- Accession codes, unique identifiers, or web links for publicly available datasets
- A description of any restrictions on data availability
- For clinical datasets or third party data, please ensure that the statement adheres to our policy

All data are provided as source data (Stock_Milutinovic_source_data.xlsx).

# Field-specific reporting

Please select the one below that is the best fit for your research. If you are not sure, read the appropriate sections before making your selection.

☐ Life sciences    ☐ Behavioural & social sciences    ☒ Ecological, evolutionary & environmental sciences

For a reference copy of the document with all sections, see nature.com/documents/nr-reporting-summary-flat.pdf

# Ecological, evolutionary & environmental sciences study design

All studies must disclose on these points even when the disclosure is negative.

| | |
|---|---|
| Study description | We performed a serial passage experiment with fungal pathogens over 10 infection cycles in ant hosts. Exposed ants were either kept alone or attended by two nestmates, to obtain two selection treatments (individual and social host), each performed in 10 replicate lines. We determined the number and identity of fungal strains remaining from the original starting mix of 6 strains for each replicate line at passages 5 and 10 (by line-level quantitative real-time PCR followed by clone-level microsatellite analysis). We compared strain diversity during the course (passage 5) and at the end of the experiment (passage 10) between the selection treatments (by Wilcoxon rank sum tests for independent samples, WRST), as well as the final composition of the strains prevailing at after the serial passage experiment under the two selection treatments (by Fisher exact test). We then characterized the 20 successful lines for their virulence (measured as induced host mortality) and transmission potential (number of spores growing out of the sporulating carcasses) in a common garden experiment (testing the main effects of selection history and current host social context and their interaction, and including replicate line and laboratory stock colony as random effects). We further determined the allogrooming intensity elicited by these fungal lines in nestmate ants (comparing the allogrooming events induced by individual and social lines by Wilcoxon rank sum tests for independent samples, WRST). Characterisation of the chemical spore profiles of the fungal lines by gas chromatography-mass spectrometry revealed lower ergosterol levels in the social lines (Permanova for overall difference between individual and social lines, Random Forest to determine important contributing compounds, WRST for the six identified important compounds, p-value adjustment for multiple testing by the Benjamini Hochberg correction). We therefore performed a bioassay applying ergosterol vs an acetone sham treatment on the ants and determined the elicited allogrooming in nestmates (comparing the allogrooming events in the two treatments by WRST). We further determined the specificity of the allogrooming-elicitation by the pure ergosterol treatment by performing a second bioassay, using the non-fungal, but chemically highly similar compound cholesterol vs sham application. |
| Research sample | As pathogen, we used 6 strains of the entomopathogenic fungus Metarhizium, all collected from the same sampling site and hence representing a natural, sympatric fungal community (3 strains of M. robertsii and 3 strains of M. brunneum), making natural coinfections of these strains likely. As host, we used workers of the Argentine ant, Linepithema humile, which are susceptible to all the six fungal strains. Queens, workers and brood were collected in 2011 and 2016 from the wide-ranging "main" supercolony that L. humile forms in large areas of Southern Europe (as a characteristic of invasive ants, nests are interconnected to supercolonies). Our sampling site was close to Sant Feliu de Guíxols, Spain (N 41° 49', E 3° 03'). The field-collected insects were brought back to the laboratory to set up large stock colonies, out of which workers were taken from outside the brood chambers for use in the experiments. |
| Sampling strategy | We ran our serial passage experiment in 10 replicate lines each of the individual and social host selection treatment, to obtain a high enough replication to be able to test for significant differences between the two treatments (as most statistical tests require a minimum of 6 replicates per treatment group). For each line at each passage, we pooled the outgrowing spores from the first 8 carcasses, which was sufficient to obtain enough spores to expose the ants for the next infection cycle without the need for additional expansion on the plate (which, due to strain-strain competition could have altered pathogen strain composition), and to freeze remaining spores for molecular analysis and clone individualisation; the latter allowed for growth of the successful lines for characterization of their virulence and transmission (in the common garden experiment), as well as their allogrooming elicitation and chemical profiles. To obtain these 8 carcasses throughout the serial passage experiment for each replicate line and passage in both of the treatments, a total of 6312 fungus-exposed ants and 8026 nestmates was used, based on preliminary assessments of mortality induction and carcass spore outgrowth. In the common garden experiment, 19-20 ants per replicate line from each of the four combinations of selection history and the current social host condition (total 797 ants) were exposed, to allow for a reliable estimation of the proportion of ants dying per replicate line and enough outgrowing carcasses for spore number quantification (median of 5 carcasses per combination, total 215). To ensure a large enough sample size for the induction of allogrooming in the 20 lines, we quantified nestmate allogrooming in three biological replicates per line. To assure reliable read-out of the gas chromatography-mass spectrometry, we ran each line in three technical replicates. As we tested for the effect of only a single compound from the natural spore profile in the bioassay, the relatively high sample size of >20 ants each receiving the ergosterol and sham treatment was chosen. The same was true for the bioassay with the non-fungal-derived cholesterol. |
| Data collection | The serial passage and common garden experiments including the molecular strain characterisations were performed as described in the study description and sampling strategy by MS and AVG. Chemical data, including running of the spore extracts in the gas chromatograph – mass spectrometer, quantitative analysis of the abundance per peak and the compound identification from the mass spectrum were obtained by FW, NK, MH and TS. The fungal line elicitation of allogrooming was observed by MN. MH performed the bioassays. |
| Timing and spatial scale | The serial passage experiment was run over a duration of ½ year in 2011-2012, followed by the molecular analysis of the strains and clones, so that the common garden experiment and the allogrooming elicitation experiment could be performed in 2013. All these experiments used the ants collected from the field in Spain in 2011 and subsequently reared in the laboratory. In 2015, the fungal lines were run on a gas chromatograph-mass spectrometer, followed by analysis of the spore profiles for their qualitative (mass spectra) and quantitative (peak integration) compound characteristics and statistical differences between the strains, revealing lower ergosterol amounts in the social than individual lines. The bioassay testing for the grooming-eliciting effect of application of pure |

ergosterol was performed in 2021 using ants sampled in 2016 during a second ant collection from the same field population in Spain. The specificity bioassay with pure cholesterol was performed in 2022 with ants collected the same year, again from the same population.

**Data exclusions**

We excluded a single value from our analysis, as we identified an outlier during the carcass spore counts (from replicate I5), which had a value that was 100-fold higher than the spore counts of the other carcasses of its replicate, suggesting a counting error in this single carcass. We tested whether exclusion of this outlier value had an effect on our analysis by rerunning the statistics including this value, which did not change the statistical outcome.

**Reproducibility**

Experiments were run in simultaneous replication, i.e. in the serial passage experiment, 10 independent replicate lines were obtained in parallel for each of the selection treatments. As our molecular analysis revealed that 7/10 individual and 6/10 social lines had the same strain as winner strain, we also performed a statistical analysis of only this subset, which showed the robustness of the effects of the individual vs social host selection treatment independent of fungal strain identity. The allogrooming elicitation was replicated three times per fungal line (3 biological replicates). In the GC-MS, each fungal line was run in 3 technical replicates, revealing high reproducibility of the method. For the bioassays, we chose a sample size of n>20 per treatment.

**Randomization**

For all experiments, workers were picked randomly from outside the brood chambers in the laboratory stock colonies, in which the ants were reared after field collection (with laboratory stock colony included as random effect in the linear models). These workers were then randomly assigned to treatment (e.g. exposed ant vs nestmate, ergosterol- / resp. cholesterol-treatment vs sham-treatment). During video acquisition of the allogrooming elicitation, the three replicates per fungal line were randomised over the cameras. For the gas-chromatography-mass spectrometry, spore extracts were run in a randomised manner, intermingled with blank runs and handling controls. In the bioassays, the treatments were randomised during video acquisition.

**Blinding**

The observers annotating the videos for allogrooming were blinded for treatment to assure bias-free observation. In detail, in the elicitation of allogrooming from fungal lines, the observer was blind for both the selection history, as well as the individual ant treatment, and in the bioassay blind for the treatment (ergosterol, respectively cholesterol, vs sham) of the treated individual.

Did the study involve field work?  ☐ Yes  ☒ No

# Reporting for specific materials, systems and methods

We require information from authors about some types of materials, experimental systems and methods used in many studies. Here, indicate whether each material, system or method listed is relevant to your study. If you are not sure if a list item applies to your research, read the appropriate section before selecting a response.

## Materials & experimental systems

| n/a | Involved in the study |
|---|---|
| ☒ | ☐ Antibodies |
| ☒ | ☐ Eukaryotic cell lines |
| ☒ | ☐ Palaeontology and archaeology |
| ☐ | ☒ Animals and other organisms |
| ☒ | ☐ Human research participants |
| ☒ | ☐ Clinical data |
| ☒ | ☐ Dual use research of concern |

## Methods

| n/a | Involved in the study |
|---|---|
| ☒ | ☐ ChIP-seq |
| ☐ | ☒ Flow cytometry |
| ☒ | ☐ MRI-based neuroimaging |

## Animals and other organisms

Policy information about studies involving animals; ARRIVE guidelines recommended for reporting animal research

**Laboratory animals**

The insects (ants) used in our experiments were collected from the field and reared in the laboratory before use in the experiments.

**Wild animals**

We used workers of the invasive Argentine ant, Linepithema humile, in our experiments. Queens, workers and brood from the main European supercolony were collected from their nests in the soil from a field population close to Sant Feliu de Guíxols, Spain (N 41° 49', E 3° 03') in 2011, 2016 and 2022. Ants were transported back to the laboratory in plastic boxes, to be reared as large stock colonies in plastic boxes containing smaller nest boxes and a plastered floor. Ants either died from fungal infection during the experiments or were frozen at the end of the experiment.

**Field-collected samples**

Stock colonies were reared in an incubator at 27 °C with day/night light cycle, which assured high productivity of the colonies as the experiments required > 15,000 workers. The experiments were carried out under conditions optimal for Metarhizium to establish infections in the ant, in a temperature- and humidity-controlled room at 23 °C, 65% RH and a 12h day/night light cycle. During experiments, ants were kept in petri dishes with plastered floor and provided with ad libitum access to a sucrose-water solution (100g/L) and plaster was watered every 2-3 days to keep humidity high.

**Ethics oversight**

We used ants (insects, invertebrates) as our study animals. The study was performed on workers of an invasive ant species, the Argentine ant, Linepithema humile. Collection of this unprotected species from the field was in compliance with international regulations, such as the Convention on Biological Diversity and the Nagoya Protocol on Access and Benefit-Sharing (ABS; permits not required for collections before 2017; 2022: ABSCH-IRCC-ES-260624-1 ESNC126 and SF0171/22). Transport to and rearing of the ants in the laboratory, as well as all experimental work followed European and Austrian law and institutional ethical guidelines of ISTA (Institute of Science and Technology Austria).

# Flow Cytometry

## Plots

Confirm that:

☐ The axis labels state the marker and fluorochrome used (e.g. CD4-FITC).

☐ The axis scales are clearly visible. Include numbers along axes only for bottom left plot of group (a 'group' is an analysis of identical markers).

☐ All plots are contour plots with outliers or pseudocolor plots.

☐ A numerical value for number of cells or percentage (with statistics) is provided.

## Methodology

| | |
|---|---|
| Sample preparation | We used flow cytometry to individualise the spores from the spore pools collected from the sporulating carcasses as described in the sampling strategy. Hence, pure spore suspensions of the harvested spores per replicate line, but no additional cell populations, constituted the starting population for flow cytometry, which were then separated into individual clones. |
| Instrument | FACS ARIA III, BD Biosciences |
| Software | Diva 6.2 software |
| Cell population abundance | Each sorted population only contained the spores of the respective fungal line. The aim of flow cytometry was not to choose a particular subpopulation, but to sort our lines as single spores into 96-well plates containing agar in each well (as an effective and reliable alternative to streaking out the spores on agar), where they grew into colony forming units (CFUs). These individual spore clones were used for (i) microsatellite analysis to obtain strain identity, and (ii) for growth expansion of the fungal lines to allow characterisation of their virulence, transmission, allogrooming elicitation and chemical profiles (growing from individual spores was required as growth of a spore pool on agar plates leads to growth inhibition of some strains due to strain-strain competition). |
| Gating strategy | The unstained spore populations were detected using the FSC (Forward Scatter) / SSC (Side Scatter) in linear mode (70 μm nozzle). Purity mode was set to 'single cell' and spore clones were obtained by sorting one particle event into each well. This procedure allowed to grow mono-clone cultivars, as spores were sorted into separate wells, and cases of spores sticking together as e.g. duplets or triplets were excluded (see Supplemental Fig. 1 as an example of the gating strategy). During method establishment, it was verified that the chosen procedure led to single spores by performing both visual checking under the microscope and by reanalysis of the sorted population. |

☒ Tick this box to confirm that a figure exemplifying the gating strategy is provided in the Supplementary Information.

