## [Peer Review File · Nature Ecology & Evolution]

Peer Review Information

Journal: Nature Ecology & Evolution

Manuscript Title: Pathogen evasion of social immunity

Corresponding author name(s): Barbara Milutinović, Sylvia Cremer

Editorial Notes:

Reviewer Comments & Decisions:

Decision Letter, initial version:

12th April 2022

Dear Professor Cremer,

Your Brief Communication, "Pathogen evasion of social immunity" has now been seen by three reviewers. You will see from their comments copied below that while they find your work of considerable potential interest, Reviewer 3 has raised substantial concerns that must be addressed in a revision before we can decide on publication in Nature Ecology & Evolution.

Reviewer 3's comments confirm our initial editorial reservations that the work is too preliminary and would benefit from additional experiments and expansion into a longer Article format. Therefore, we agree with Reviewer 3 that additional experiments should be added to make a stronger case for generality, i.e. showing that the results hold for different strains. We will overrule Reviewer 3 on the suggestion to somehow manipulate the fungus directly to produce ergosterol, as that is outside the scope of the study; however, a better case should be made that the amount of ergosterol applied to the ants is comparable to what would be produced by the fungus.

* Include a "Response to reviewers" document detailing, point-by-point, how you addressed each referee comment. If no action was taken to address a point, you must provide a compelling argument. This response will be sent back to the referees along with the revised manuscript.

* If you have not done so already we suggest that you begin to revise your manuscript so that it conforms to our Article format instructions at <http://www.nature.com/natecolevol/info/final-submission>. Refer also to any guidelines provided in this letter. Note that we are recommending a change from Brief Communication to Article, so you may take advantage of longer word limits, display item counts, etc.

* Include a revised version of any required reporting checklist. It will be available to referees (and,

2potentially, statisticians) to aid in their evaluation if the manuscript goes back for peer review. A revised checklist is essential for re-review of the paper.

[REDACTED]

If you wish to submit a suitably revised manuscript we would hope to receive it within 6 months. If you cannot send it within this time, please let us know. We will be happy to consider your revision so long as nothing similar has been accepted for publication at Nature Ecology & Evolution or published elsewhere.

Nature Ecology & Evolution is committed to improving transparency in authorship. As part of our efforts in this direction, we are now requesting that all authors identified as 'corresponding author' on published papers create and link their Open Researcher and Contributor Identifier (ORCID) with their account on the Manuscript Tracking System (MTS), prior to acceptance. This applies to primary research papers only. ORCID helps the scientific community achieve unambiguous attribution of all scholarly contributions. You can create and link your ORCID from the home page of the MTS by clicking on 'Modify my Springer Nature account'. For more information please visit www.springernature.com/orcid.

Thank you for the opportunity to review your work.

[REDACTED]

Reviewer expertise:

Reviewer #1: Chemical ecology

Reviewer #2: Ant-fungus interactions

Reviewer #3: Ant-fungus interactions

Reviewers' comments:

Reviewer #1 (Remarks to the Author):

2I really appreciate this short and lively manuscript on the outcome of the selective pressures imposed by host grooming and social care on pathogenic ant fungi.

The authors tested the outcome of selection in terms of pathogen virulence and transmission but also investigated the chemical profile of the fungal spores and how it evolved after the repeated infection cycles, and identified (and tested) the chemical cue triggering sanitary caregiving by host ants.

I only have very minor comments, which are often simple suggestions.

Abstract:

Maybe highlight that you got experimental evidence.

Main text:

L25 (or ~L10 in online methods): some more details about the collection would be welcome. Although Giraud et al (2002, PNAS) highlighted the lack of genetic variation of *Linepithema humile* in Southern Europe, they also showed that two supercolonies exist and that they are aggressive to each other. How likely you are to have collected from only one? Even if you collected ants from only one supercolony, how did you keep "colonies" separate in the lab? The level of physical "separation" between colonies in the lab might affect the level of independence between data collected from the different replicates. In other words, my question is: how separate are the 10+10 replicates in your study? Is there any genetic or environmental-induced variation? If not, in which way the lack of any variation in the ant population might have affected your results?

Page 3, Line 10: at what extent is the prevailing presence of a single fungal strain associated with the lack of variation in host genotype?

Page 4, Lines 2-12: virulence and transmission of pathogens are introduced here, but defined precisely only at page 15, line 23-24 in online methods. I suggest to clearly define these measures the first time they are mentioned both in the main text and in the material and methods.

Page 6, Line 2: "the specific compounds that stimulate these responses are largely unknown" I misinterpreted the sentence understanding that you were interested in cuticular compounds/pheromones produced by ants that trigger social grooming. Maybe better to specify you are interested in spore compounds? Btw, is grooming and especially allogrooming there only after spores are there?

Page 6, Line 16: "PAMP": is this acronym useful? It appears only twice in the main text...

Page 7, Line 2: How would you exclude that the decrease in ergosterol is the result of the selective pressure imposed by ant sanitation behaviors? (ergosterol-triggered grooming selects for lower ergosterol levels in fungal spores).

Maria Cristina Lorenzi

Reviewer #2 (Remarks to the Author):

3The research presented in this manuscript investigates how fungal pathogens of the genus *Metarhizium* respond to selection pressure in the form of social grooming practices by nestmates of infected *Linepithema humile* ants. This is a highly relevant research question that has been understudied so far and warrants detailed investigation. The research that is being presented in this manuscript is a well-rounded approach that results in well-supported, clear results and even goes as far as to expose a plausible mechanism through which ants might recognize fungal spores. In response to selection pressure through conspecific grooming, fungal spore production increased over as little as 10 generations. The spores were less virulent but also less detectable. GC-MS and follow-up exposure experiments demonstrated that changed ergosterol levels played a role in these findings. Overall, I am of the opinion that this manuscript presents a well-crafted, well-rounded, and thoroughly thoughtfully designed approach to address a highly relevant question. As such, this work represents a significant contribution to the literature that is of interest to the broad audience of Nature Ecology and Evolution. The manuscript is also well-written, easy to follow, and in my opinion all data is well-presented and the methods have been thoroughly laid out. As far as I can judge, I also agree with the statistics performed to analyze the data. As such, I think this work is acceptable for publication.

The only very minor contribution that I can make to improve this manuscript is to suggest to revisit the Allogrooming elicitation by the fungal lines section in the Methods. This section suffers from some minor grammar and language flaws that could use some reconsidering to improve its readability. Other than this very, very minor insignificant comment I have nothing to contribute and want to commend the authors for this beautiful work.

Reviewer #3 (Remarks to the Author):

The authors treat ants with a mix of six entomopathogenic fungal strains, and then follow the infection either in a social (with two other ants that groom the infected individual) or an individual setting. After ten passages, they find that usually one of the six strains prevails, and that the strains that have undergone selection in the social setting show decreased virulence and increased spore production compared to strains from the individual setting, on average. This is interpreted as being reflective of a tradeoff between virulence and spore production, and selection to evolve higher spore production in the social setting where the ants remove spores via grooming. The authors also find that socially selected strains elicit less allogrooming, which is associated with a reduction in ergosterol and a few other compounds. That ergosterol might be directly involved in inducing allogrooming is supported by an experiment in which the authors apply the compound directly to ants and observe increased grooming. This demonstration of how social immunity in ants affects pathogen evolution is novel and interesting, and the experiments presented here are overall well conducted and analyzed (data presentation and statistical analyses also meet the journal guidelines as far as I can tell). However, I felt that some of the main conclusions could be substantially bolstered with additional experiments, and I would encourage the authors to fill in these gaps.

Comments related to Figure 1:

One main concern is that the authors use a mix of different strains as their starting condition, but ultimately test effects on (mostly) individual strains at the end of the experiment. This confounds the effects of inter-strain competition with the effects of social vs. individual immunity, and leads to

4uncontrolled variation between replicates in terms of strain trajectories and endpoints. One of their strains, R1, disproportionately outcompetes all other strains in both their conditions, suggesting that something is special about this strain. This calls into question how generalizable their results are. I was unable to find a convincing justification for why the authors chose this setup as their initial starting point, rather than conducting the experiment with a few individual strains separately, which would have been more easily interpretable. As the authors are aware (p. 4 line 5), their design makes it impossible to compare the endpoint of the experiment directly to the start point. While I agree that the experiment suggests that social immunity has a selective effect on virulence and spore production in R1, it is not clear whether this is a direct effect, or whether it is contingent on some unknown interaction between different fungal strains. To draw robust conclusions, I suggest that the authors perform additional experiments in which they compare single strain performance before and after the two different selective regimes for at least a few of their strains.

A related point on p. 4 (starting on line 24): the conclusion that the findings are independent of successful strain identity is not justified. In the majority of cases, R1 is the dominant strain, and the results will be mostly driven by R1 in the combined analysis. The authors would have to show that they obtain similar results for strains other than R1, something for which they currently don't have the data. However, as I mention above, I think it would be important to conduct this experiment with a few different strains individually as the starting condition. This would then also allow the authors to draw conclusions about the generality of the observed patterns across strains.

Comments related to Figure 2:

The identification of reduced ergosterol in fungi having undergone selection in a social setting, and the demonstration that ergosterol may elicit grooming, are very interesting. However, the authors demonstrate the effect of ergosterol on allogrooming not by manipulating the compound on the fungal spores directly, but by applying it to ants. This again seemed like an odd choice. For once, I wasn't able to figure out whether the amount of ergosterol experimentally applied to ants mirrors the difference found on spores subjected to different selective regimes, i.e., whether it is biologically relevant. The response to ergosterol treatment is not very strong (Figure 2b), and generally an order of magnitude lower than what the authors see in the treatments with actual fungal spores (Figure 2a vs 2b). So this is important to get right, and it should be clarified. The authors also only compare their ergosterol treatment to a solvent control (acetone). However, that doesn't necessarily mean that ergosterol specifically elicits allogrooming – wouldn't most compounds when applied to the ant cuticle in sufficient quantity elicit allogrooming? What the authors should really show here is that an increase in applied ergosterol quantities similar to the change observed in fungal strains increases allogrooming. Ideally, the authors would of course experimentally alter ergosterol levels in fungus spores directly (can this be done genetically or pharmacologically?), then infect ants, and measure allogrooming by nestmates.

A few additional thoughts on that experiment:

- The authors often talk about altered "abundance" of compounds, but as far as I can tell, what they are really measuring is "relative abundance". I would be more explicit about this throughout the text. Without data on changes in absolute abundance, it might be difficult to determine biologically relevant quantities for experiments.
- On page 6, line 16, I would write "a sufficient pattern" – you can't show it's the only one. But see my general concerns re this interpretation above.
- Page 6, line 23: I couldn't find an explanation of why pathogen-pathogen competition should be more intense under social immunity in the current manuscript (although I didn't check the paper cited

5here). Wouldn't that go against your finding that the strain-level selective outcomes are actually very similar across your two treatments (Fig. S1)?

Author Rebuttal to Initial commentsComments by Reviewer #1:

I really appreciate this short and lively manuscript on the outcome of the selective pressures imposed by host grooming and social care on pathogenic ant fungi.

The authors tested the outcome of selection in terms of pathogen virulence and transmission but also investigated the chemical profile of the fungal spores and how it evolved after the repeated infection cycles, and identified (and tested) the chemical cue triggering sanitary caregiving by host ants.

I only have very minor comments, which are often simple suggestions.

Reply: Thank you very much for the appreciation of our work. We changed the manuscript according to your suggestions, as detailed below.

R 1.1. Abstract:

Maybe highlight that you got experimental evidence.

Reply: We have rewritten the abstract (now also allowing for 200 words due to the invitation to expand to an article) and included our experimental evidence.

R 1.2. Main text:

*L25 (or ~L10 in online methods): some more details about the collection would be welcome. Although Giraud et al (2002, PNAS) highlighted the lack of genetic variation of *Linepithema humile* in Southern Europe, they also showed that two supercolonies exist and that they are aggressive to each other. How likely you are to have collected from only one? Even if you collected ants from only one supercolony, how did you keep “colonies” separate in the lab? The level of physical “separation” between colonies in the lab might affect the level of independence between data collected from the different replicates. In other words, my question is: how separate are the 10+10 replicates in your study? Is there any genetic or environmental-induced variation? If not, in which way the lack of any variation in the ant population might have affected your results?*

Reply: All our ants stem from the “main supercolony” of the Argentine ant (Milutinovic et al 2020) and were reared in large, stable stock colonies with very high queen and worker numbers. We chose the Argentine ant system particularly because of the low genetic variation in this invasive ant species, such that this uniformity in the host allowed a more standardized selection in the fungus. We also kept the rearing and experimental conditions constant throughout the experiment. At every infection cycle, we took ants from the stock and used them for the experimental exposures. We assured that the 20 fungal lines were strictly separated, with no exchange between them, by keeping workers after exposure in

separate petri dishes (alone in the individual and with its nestmates in the social treatment) until they died and produced a sporulating cadaver. We then only pooled the spores of the cadavers from the same replicate line to expose the next host cycle of that line. Therefore, any difference between the selection lines can be attributed to the difference between individual vs social host condition, but not to difference in host genotype. We have expanded the explanation of our experimental design in the manuscript (p. 4, l. 21 ff) and our methods sections “Ant host” (p. 11, l. 13 ff) and “Serial passage experiment” (p. 13, l. 7 ff) to add this additional information.

R 1.3. Page 3, Line 10: at what extent is the prevailing presence of a single fungal strain associated with the lack of variation in host genotype?

Reply: We are not sure there is enough data to answer this question comprehensively, as – to our knowledge – this is the first study examining the long-term effects of social immunity on pathogen-pathogen competition. Given, however, that many studies in the field of host-pathogen interactions find an effect of both host and pathogen genotype, it is to be expected, that, had we instead used e.g. the Catalan supercolony, or a different ant species, the strain sorting may have yielded different prevailing *Metarhizium* strains at the end of our experiment (note that the reduction of strain number per se is a very robust finding, occurring across selection regimes, but being slower under social immunity). Notably, however – and this is a point that we missed to explain sufficiently in the previous version of the manuscript – our findings on the effect of social immunity vs individual immunity only, are indeed robust across strains in our study (see also below response to R3.1.). We now provide these analyses throughout the results section.

R 1.4. Page 4, Lines 2-12: virulence and transmission of pathogens are introduced here, but defined precisely only at page 15, line 23-24 in online methods. I suggest to clearly define these measures the first time they are mentioned both in the main text and in the material and methods.

Reply: We now provide the clear definitions of how we measured both virulence and transmission at first time mention in both the main manuscript and the methods.

R 1.5. Page 6, Line 2: “the specific compounds that stimulate these responses are largely unknown” I misinterpreted the sentence understanding that you were interested in cuticular compounds/pheromones produced by ants that trigger social grooming. Maybe better to specify you are interested in spore compounds? Btw, is grooming and especially allogrooming there only after spores are there?

Reply: We have removed this ambiguity and have rewritten this section (p. 7, l. 20 ff). We now clearly mention that we were interested in spore compounds, and stating the relevance of both, pathogen- and host-derived cues.

There is indeed some baseline grooming also in untreated ant colonies, which will contribute to non-sanitary functions (such as formation of the colony gestalt odour). Yet, these baseline grooming data of completely untreated individuals is rarely obtained in studies interested in pathogen-induced sanitary care behavior, as the appropriate control for these studies is a sham control, in which the hosts are treated by a non-pathogenic sham treatment, resp. handling control. Therefore, non-sanitary baseline grooming is an inclusive part of the sham treatment and sanitary grooming induction can be quantified as the difference to this overall sham grooming. To make this distinction clearer, we changed the previously vague wording “these responses” by “sanitary responses”.

R 1.6. Page 6, Line 16: "PAMP": is this acronym useful? It appears only twice in the main text...

Reply: We have now discarded the acronym PAMP, and instead use MAMP, which we believe is a more accurate acronym since ergosterol is found in all fungi, also the ones that are not pathogens. As we now expand in the manuscript (p. 10, l. 17 ff), employing such a high gain-low cost sanitary response like grooming to all detected fungi, even if only some of them can cause host disease, will be very beneficial to the ants.

MAMPS are a well-known concept in triggering an immune response of the physiological immune system, and the observations that pathogens can reduce their MAMPS to evade the physiological immune response is an important parallel, as – to our knowledge – our study is the first to show that pathogens can reduce their MAMPS in response to social immunity. We therefore think that the use of this term helps readers with a main interest in immunology to connect to our work and therefore enhances access of the broad readership to our work. We think that such interdisciplinary discourse is beneficial given the overarching common principles of organismal immunity within bodies and social immunity in social insect colonies, and therefore also included it in the abstract.

R 1.7. Page 7, Line 2: How would you exclude that the decrease in ergosterol is the result of the selective pressure imposed by ant sanitation behaviors? (ergosterol-triggered grooming selects for lower ergosterol levels in fungal spores).

Maria Cristina Lorenzi

Reply: The reviewer is correct, we cannot and do not want to exclude this; in fact – as we expand more clearly in the revised version of the manuscript (p. 9, l. 23 ff) – ergosterol-triggered grooming likely explains why the ergosterol-reduced spores will get a selective benefit. In this manuscript section, our aim was to speculate on a scenario suggesting a possible mechanism that could have sparked the reduction in ergosterol levels in the social lines in the first place, such as resource limitation (because of increased spore production, less nutrients are left for investment into other components). Notably, we started our experiment with mono-clonal strains, such that e.g. all *M. robertsii* R1 winners started with exactly the same ergosterol level and all differences between the social and individual R1 lines at the end of the experiment were the result of different selection.

Comments by Reviewer #2:

The research presented in this manuscript investigates how fungal pathogens of the genus Metarhizium respond to selection pressure in the form of social grooming practices by nestmates of infected Linepithema humile ants. This is a highly relevant research question that has been understudied so far and warrants detailed investigation. The research that is being presented in this manuscript is a well-rounded approach that results in well-supported, clear results and even goes as far as to expose a plausible mechanism through which ants might recognize fungal spores. In response to selection pressure through conspecific grooming, fungal spore production increased over as little as 10 generations. The spores were less virulent but also less detectable. GC-MS and follow-up exposure experiments demonstrated that changed ergosterol levels played a role in these findings. Overall, I am of the opinion that this manuscript presents a well-crafted, well-rounded, and thoroughly thoughtfully designed approach to address a highly relevant question. As such, this work represents a significant contribution to the literature that is of interest to the broad audience

of *Nature Ecology and Evolution*. The manuscript is also well-written, easy to follow, and in my opinion all data is well-presented and the methods have been thoroughly laid out. As far as I can judge, I also agree with the statistics performed to analyze the data. As such, I think this work is acceptable for publication.

Reply: We thank the reviewer for this very positive evaluation of our work.

R 2.1. The only very minor contribution that I can make to improve this manuscript is to suggest to revisit the Allogrooming elicitation by the fungal lines section in the Methods. This section suffers from some minor grammar and language flaws that could use some reconsidering to improve its readability. Other than this very, very minor insignificant comment I have nothing to contribute and want to commend the authors for this beautiful work.

Reply: We have changed the respective text section to make it more readable (p. 19, l. 8 ff).

Comments by Reviewer #3:

The authors treat ants with a mix of six entomopathogenic fungal strains, and then follow the infection either in a social (with two other ants that groom the infected individual) or an individual setting. After ten passages, they find that usually one of the six strains prevails, and that the strains that have undergone selection in the social setting show decreased virulence and increased spore production compared to strains from the individual setting, on average. This is interpreted as being reflective of a tradeoff between virulence and spore production, and selection to evolve higher spore production in the social setting where the ants remove spores via grooming. The authors also find that socially selected strains elicit less allogrooming, which is associated with a reduction in ergosterol and a few other compounds. That ergosterol might be directly involved in inducing allogrooming is supported by an experiment in which the authors apply the compound directly to ants and observe increased grooming. This demonstration of how social immunity in ants affects pathogen evolution is novel and interesting, and the experiments presented here are overall well conducted and analyzed (data presentation and statistical analyses also meet the journal guidelines as far as I can tell). However, I felt that some of the main conclusions could be substantially bolstered with additional experiments, and I would encourage the authors to fill in these gaps.

Reply: We thank the reviewer for the thoughtful comments, which helped us to substantiate our work. As detailed below, we addressed all raised points by a combination of additional experimental work, more in-depth data analysis, and detailed descriptions (as by editorial invitation, the manuscript format has now changed to a longer article). We think that the revised version now much more coherently describes our eco-evolutionary approach, provides evidence for the generality of our findings and sets the testing with the pure ergosterol compound into clearer context. We also expanded on the discussion of how we expect selection by social immunity to affect pathogen competition in our study system.

R 3.1. Comments related to Figure 1:

One main concern is that the authors use a mix of different strains as their starting condition, but ultimately test effects on (mostly) individual strains at the end of the experiment. This confounds the effects of inter-strain competition with the effects of social vs. individual immunity, and leads to uncontrolled variation between replicates in terms of strain trajectories and endpoints. One of their strains, R1, disproportionately outcompetes all other

strains in both their conditions, suggesting that something is special about this strain. This calls into question how generalizable their results are. I was unable to find a convincing justification for why the authors chose this setup as their initial starting point, rather than conducting the experiment with a few individual strains separately, which would have been more easily interpretable. As the authors are aware (p. 4 line 5), their design makes it impossible to compare the endpoint of the experiment directly to the start point. While I agree that the experiment suggests that social immunity has a selective effect on virulence and spore production in R1, it is not clear whether this is a direct effect, or whether it is contingent on some unknown interaction between different fungal strains. To draw robust conclusions, I suggest that the authors perform additional experiments in which they compare single strain performance before and after the two different selective regimes for at least a few of their strains.

A related point on p. 4 (starting on line 24): the conclusion that the findings are independent of successful strain identity is not justified. In the majority of cases, R1 is the dominant strain, and the results will be mostly driven by R1 in the combined analysis. The authors would have to show that they obtain similar results for strains other than R1, something for which they currently don't have the data. However, as I mention above, I think it would be important to conduct this experiment with a few different strains individually as the starting condition. This would then also allow the authors to draw conclusions about the generality of the observed patterns across strains.

Reply: We agree with all the points raised by the reviewer and are sorry that our previous manuscript version did not give enough background to follow our reasoning for the experimental design. We now expanded the introduction (p. 3, l. 18 – p. 4, l. 18) to make this clearer.

One interesting effect of grooming is not only that it reduces (absolute) spore number for all pathogens, but that it does so to a different degree for different pathogens depending on their infection properties, thereby also changing the (relative) outcome when spores of multiple strains are in competition. In particular, our previous work on the same study system had revealed that, during a single host infection cycle, grooming by the Argentine ant shifts the competitive outcome between *Metarhizium* species in favour of *M. brunneum* over the otherwise more competitive *M. robertsii* (Milutinovic et al 2020 *Ecol Lett*). With the current study, we aimed at understanding the long-term consequences that this bias-introducing effect has on the composition of the pathogen community and strain fitness. It was therefore absolutely necessary for us to start the experiment with co-infections. Using coinfections, which are in fact more prevalent in nature than single-strain infections (Balmer & Tanner 2011 *Lancet Inf. Dis.*), our aim was to not only test the direct effect of reduced infection likelihood via social grooming, but also the indirect effects of how social immunity on the long term modulates the pathogen community – and hence disease ecology.

We are fully aware that this experimental design introduced variation by different strain success during, and at the end of the ten host cycles. However, finding out the dynamics of strain sorting over multiple infection cycles was exactly one of our main study questions. Indeed, *M. robertsii* strain R1 turned out as the universal winner of the competition (across selection treatments); whilst this is consistent with its high competitive ability in a single host infection cycle (Milutinovic et al 2020 *Ecol Lett*), its marked dominance in our long-term experiment could not have been predicted. However, this prominence of R1 in 7 individual and 6 social lines, allowed us to in fact gain statistical power and test how the effect of social immunity altered the same starting strain (started from a mono-clonal culture) in these 13 independent selection lines, clearly showing that selection pressure of social immunity

consistently induced lower virulence, increased transmission potential and social immunity evasion.

We agree that this also means that the other successful strains cannot be compared between individual and social selection with a proper statistical analysis, and have thus also revised our previous formulation that our conclusions are independent of strain identity. Still, *M. robertsii* strain R3 won in 3 individual and 2 social selection lines. Descriptive statistics comparing the individual and social selection lines unambiguously show the exact same pattern (information that we have now added in pages 7, l. 2; l. 15 and p. 8, l. 5ff). The *M. brunneum* B2 strain won in 2 social selection lines, but in none of the individual lines. This strongly suggests that social immunity can shape pathogen communities and give even a long-term advantage to an otherwise weak competitor (it won the competition in our experiment in 2/10 social replicate lines, which is still large contribution to its fitness). Since it didn't prevail in any of the individual lines, we cannot make a direct comparison of the competitive ability of *M. brunneum* B2 selected under the different treatments, yet it is notable that the two B2 social lines behave very similar to the social R1 and R3 lines.

Overall, this reveals that the statistically significant effects within R1 can be generalized (with admittedly low statistical power) also to other winner strains in our experiments. We hope that our study triggers further work testing to evaluate generality of the selective effects of social immunity in other host pathogen systems and populations.

Indeed, we cannot compare the start point to the endpoint due to the difference in strain numbers, but can only compare between the endpoints of the social and the individual selection lines (that all had single strain dominance), which even gives statistically testable results for our main strain. We think that even in an experiment in which one would not be interested in the modulatory effect of social immunity on pathogen-pathogen competition (outlined above), but only on the effect of social immunity on individual starting strains (as suggested as an additional experiment by the reviewer), the comparison start to end would still not be the relevant comparison to test for the effect of social immunity. This is because it is impossible to only induce social immunity in the absence of individual immunity, and any start-to-end comparisons would always also comprise selection by individual immunity as well. Therefore – independent of starting point – to derive the pure effect of social immunity per se, the only approach we see appropriate is to induce selection of (i) individual immunity only and (ii) individual plus social immunity and then compare the end points of the two.

As the reviewer further points out, our design does not allow to disentangle whether the lower ergosterol levels in the social lines are the result of a direct selection on this trait, or whether it may result from some interaction between the strains. We do fully agree (see also our reply to R1.7.), and further think that those indirect effects may even have introduced a lower ergosterol level e.g. in the social R1 lines than the individual R1 lines, from the same starting clone, possibly due to ergosterol limitation when producing double the number of spores per social than individual cadaver. This may be a mechanism causing a difference that selection by grooming could then have acted upon, giving further benefit to low-ergosterol lines, as they could evade allogrooming. We have expanded the section in the discussion where we outlined this evolutionary scenario (p. 11, l. 2 ff).

We are sorry that our study design and incentive had not become clear in the previous manuscript version, and hope to now provide all the necessary background revealing that our study addresses not only evolutionary but the evolutionary-ecological effects of social host defences in a natural pathogen community.

R 3.2. Comments related to Figure 2:

The identification of reduced ergosterol in fungi having undergone selection in a social setting, and the demonstration that ergosterol may elicit grooming, are very interesting. However, the authors demonstrate the effect of ergosterol on allogrooming not by manipulating the compound on the fungal spores directly, but by applying it to ants. This again seemed like an odd choice. For once, I wasn't able to figure out whether the amount of ergosterol experimentally applied to ants mirrors the difference found on spores subjected to different selective regimes, i.e., whether it is biologically relevant. The response to ergosterol treatment is not very strong (Figure 2b), and generally an order of magnitude lower than what the authors see in the treatments with actual fungal spores (Figure 2a vs 2b). So this is important to get right, and it should be clarified. The authors also only compare their ergosterol treatment to a solvent control (acetone). However, that doesn't necessarily mean that ergosterol specifically elicits allogrooming – wouldn't most compounds when applied to the ant cuticle in sufficient quantity elicit allogrooming? What the authors should really show here is that an increase in applied ergosterol quantities similar to the change observed in fungal strains increases allogrooming. Ideally, the authors would of course experimentally alter ergosterol levels in fungus spores directly (can this be done genetically or pharmacologically?), then infect ants, and measure allogrooming by nestmates.

Reply: We agree with the reviewer that indeed it would have been an elegant experiment to manipulate spore ergosterol levels, e.g. via genetic modification. However, studies that have tried this with other fungi revealed that such manipulations had far-reaching effects on other spore traits, like causing deformed conidia (Liu et al 2013 *Mol Plant Pathol*), thereby preventing them from being a control that would only deviate from the healthy spores by their ergosterol levels. Also application of ergosterol to the spores in suspension is not a promising alternative as ergosterol is solvable in alcohols, like acetone, but not in the surfactants needed to get spores into suspension. We therefore thank the editor for not requiring spore ergosterol level changes for this revision.

However, practical considerations were not the decisive criteria why we chose to apply the pure ergosterol compound on the ants and not the spores. Instead, this was because we wanted to test if ergosterol by itself was able to induce sanitary grooming, or whether it may only be a grooming enhancer, needing to act in synergy with the other chemical spore compounds, or even the tactile cues that the spores will provide to their host. It is known that social insects often react only to relative changes of different compounds in bouquet compositions, rather than to individual compounds, as we can show here. This, we think, gives our finding particular relevance, as it qualifies ergosterol *per se* as a microbe-associated molecular pattern (MAMP) that in itself is sufficient to induce social immunity.

We would disagree with the reviewer that the grooming in the bioassay is one order of magnitude lower than what we observed elicited by the selected spores. We had only given this information in the methods, but the allogrooming frequency reported in panel a reflected the sum of allogrooming in our 3 biological replicates per line, i.e. in 90 instead of 30 min. Therefore, the median of 23.5 elicited grooming events by the social line spores in 90 min (i.e. 7-8 per 30 min), actually lies in a comparable range than the median of 5.5 in the 30 min of the bioassay. However, absolute grooming numbers cannot reliably be compared across experiments, as they can vary a lot depending on many factors, including that the ants are e.g. not in the laboratory for the same duration, differ slightly in their food intake (which is known to affect their cuticular hydrocarbon patterns and hence overall smell) etc. In our experiments, we even used different application modes (the surfactant TritonX for the spores,

and acetone for the ergosterol). Therefore, we have modified our figure to prevent unsubstantiated visual suggestions of comparability.

We have now established novel methodology to quantify how many spore-equivalents of ergosterol we sprayed on in our bioassay (detailed in the new methods section “Allogrooming elicitation by pure ergosterol” pages 23-27. To this end, we compared the sprayed-on ergosterol abundance to the ergosterol abundance of ants exposed to an average of 10^3 spores (as absolutely quantified by droplet digital PCR). The ergosterol levels applied during the bioassay equals the ergosterol present in approx. 3×10^5 spores. Since insect cadavers – both *Linepithema* ants and prey insects, which the ants will scavenge for food – typically carry 10^6 to 10^9 spores/cadaver, our applied dose was therefore well in the range of a natural exposure dose.

We also experimentally addressed the concern by the reviewer that any compound applied in this amount would elicit grooming. Whilst unspecific grooming is known to be elicited by non-pathogenic tactile cues (like talcum powder or ash; Zhukovskaya et al 2013 *Insects*), individual compounds that are applied via a quickly-evaporating solvent should not necessarily induce such unspecific grooming. We tested this by repeating our bioassay with the animal membrane component cholesterol, a compound that is structurally-similar to ergosterol, but not of fungal origin, thereby not qualifying as a possible cue for fungal infection. In contrast to the ergosterol bioassay, application of cholesterol (also solved in acetone and sprayed on the ants) did not lead to any significant change to the baseline control (if anything, grooming towards cholesterol-sprayed ants was rather slightly reduced; new figure 3d). Even if this experiment does not exclude that also some further compounds may elicit grooming when applied (in fact, we would expect that the ants should react also to other compounds that they could link to a possible pathogen threat), it demonstrates that the results of our bioassay were not just caused by some unspecific effects of e.g. sterol applications.

The valuable points made by the reviewer raised our awareness of the limitations of this approach, and we think that both the added estimate of spore-equivalents applied in the bioassay and the cholesterol experiment testing for the specificity of the ants' grooming induction, have strongly substantiated our conclusions.

A few additional thoughts on that experiment:

R 3.3. - *The authors often talk about altered “abundance” of compounds, but as far as I can tell, what they are really measuring is “relative abundance”. I would be more explicit about this throughout the text. Without data on changes in absolute abundance, it might be difficult to determine biologically relevant quantities for experiments.*

Reply: We are sorry about the confusion created by our terminology. We included multiple internal standards (spanning the whole range of the retention times of the samples) per sample, as a standard procedure to control for potential small differences e.g. in injection volume across samples. For each compound, we then calculate its abundance in comparison to the closest internal standard, i.e. its “internal standard response factor (ISTD response factor)”. Our measure thus represents a standardized, yet absolute measure, and not the relative proportion of the respective compound on the overall bouquet (where one would sum the ISTD response factors of all the 40 compounds, set them to 100% and then calculate the proportion of each compound on the overall bouquet). Therefore, our terminology of relative abundance (relative, to ISTD) deviated from the common usage in the field (relative to other compounds) and was hence confusing. We could therefore indeed measure the absolute ergosterol abundance. We have now changed this from relative abundance to abundance in

the figure 3b and explain that we give it as ISTD response factor (which is further explained in the methods).

R 3.4. - On page 6, line 16, I would write "a sufficient pattern" – you can't show it's the only one. But see my general concerns re this interpretation above.

Reply: We have changed this section accordingly.

R 3.5. - Page 6, line 23: I couldn't find an explanation of why pathogen-pathogen competition should be more intense under social immunity in the current manuscript (although I didn't check the paper cited here). Wouldn't that go against your finding that the strain-level selective outcomes are actually very similar across your two treatments (Fig. S1)?

Reply: We have markedly altered the manuscript by giving more background on this (as detailed above). Figure S1 (now Figure 1b of the main manuscript) reveals that the pathogen diversity remains higher for a longer period in the social than in the individual selection treatment, which is driven by the effect of grooming giving a higher competitive advantage to the otherwise poor competitors of *M. brunneum*. It is known that competition promotes spore production, hence, even if strain sorting often led to R1 or R3 prevailing after endured competition in both selection treatments, as *M. robertsii* is, after all the better competitor, the strains have undergone a different selection experience in the socially- than individually-selected lines, with the same R1 and R3 strain having experienced much higher and longer-endured competition than the lines emerging from individual hosts. We now explain this more clearly in the revised manuscript p. 5, l. 21 ff and p. 9, l. 14 ff.

Decision Letter, first revision:

8th December 2022

Dear Dr. Cremer,

Thank you for submitting your revised manuscript "Pathogen evasion of social immunity" (NATECOLEVOL-220315974A). It has now been seen again by the original reviewers and their comments are below. The reviewers find that the paper has improved in revision, and therefore we'll be happy in principle to publish it in Nature Ecology & Evolution, pending minor revisions to satisfy the reviewers' final requests and to comply with our editorial and formatting guidelines.

[REDACTED]

Reviewer #1 (Remarks to the Author):

I appreciate this work and the manuscript is well written. The Authors have also satisfactorily replied to questions and expanded the text where needed.

Reviewer #3 (Remarks to the Author):

I appreciate the authors' efforts to revise the manuscript, including the additional experiments related to their findings on ergosterol. At the same time, they decided not to follow my suggestion to include an experiment in which individual fungus strains are subject to selection. I still think that this would have been a useful addition and would have helped clarify some of the current results (see below). At the same time, I understand that these experiments are laborious and require time and effort. The other two reviewers did not seem to be concerned about this, and one could arguably make the case that the study's findings in their current form are of sufficient interest.

Some additional thoughts on the authors' response to my initial comments related to Figure 1: If I understand correctly, the core argument of the authors here is that studying co-infecting strains is crucial because it more closely approximates natural infections, and because they previously found that allogrooming alters the relative abundance of strains, which makes coinfections more interesting. As to the first point, that might be true, but I'm not sure. They cite Balmer & Tanner 2011 in their

16response, but in the abstract that paper says "multiple-strain infections usually reach considerable frequencies (median 11.3%, mean 21.7% of infections)..." That's not exactly evidence that coinfections are more prevalent than single-strain infections, as claimed by the authors. Rather, coinfections occur and aren't that rare. It also seems that the proportion of coinfections is quite variable across pathogens, so the relevant question here is how frequent natural coinfections of *Metarhizium* are in ants. I couldn't find data on this (admittedly I only did a very cursory search), and my impression is that this is an open question. As to the second point, I agree that the dynamics in mixed-strain infections are interesting. But that doesn't get around the issues that a) it's hard to extrapolate from a single mix and b) you lose some of the information you'd get from single-strain infections. The authors' argument about comparing start and endpoints seems valid, but it would still be good to know what happens with single-strain infections in their experimental design. Maybe something for future work?

Additional comments (mostly minor, but the issue of reduced strain diversity needs to be clarified):

Line 39: omit "(MAMP)"

Line 79: this is the Balmer & Tanner 2011 paper. The authors cite it as showing that "coinfections naturally occur more frequently than single pathogen infections", but I don't think that's a fair representation of that paper (see above).

Line 80: omit comma after "to"

Line 97: This title seems to contradict the findings in this section: both social and individual immunity reduce the diversity in the pathogen community (rather than maintaining it), and there's in fact no difference between the two treatments in this effect (line 934).

Lines 112-114. This is the one statement that seems to be aligned with the section title, but the authors give no statistical comparison or refer to a table or figure that would show this. I assume they're referring to the data after 5 passages here? But I can't see where this difference is quantified.

Lines 153 to 157: I'd suggest a few edits to this sentence so it reads: "Whilst statistical testing is only possible for this prominent R1 strain, the three individual and two social lines in which *M. robertsii* R3 persisted showed similar patterns, with the social lines inducing 75% of the mortality but producing 340% of the spore outgrowth of the individual lines." Also, in the subsequent sentence, I would change "This shows" to "This suggests", simply because you can't do statistical comparisons for R3.

Line 167: omit "even"

Lines 206-208: See above – I can't see where the evidence for this is. As far as I can tell, there is no statistical test that would corroborate this conclusion, and the figure legend on line 934 seems to report the contrary. This needs to be clarified.

Our ref: NATECOLEVOL-220315974A

1716th December 2022

Dear Dr. Cremer,

Thank you for your patience as we've prepared the guidelines for final submission of your Nature Ecology & Evolution manuscript, "Pathogen evasion of social immunity" (NATECOLEVOL-220315974A). Please carefully follow the step-by-step instructions provided in the attached file, and add a response in each row of the table to indicate the changes that you have made. Please also check and comment on any additional marked-up edits we have proposed within the text. Ensuring that each point is addressed will help to ensure that your revised manuscript can be swiftly handed over to our production team.

****We would like to start working on your revised paper, with all of the requested files and forms, as soon as possible (preferably within two weeks). Please get in contact with us immediately if you anticipate it taking more than two weeks to submit these revised files.****

In recognition of the time and expertise our reviewers provide to Nature Ecology & Evolution's editorial process, we would like to formally acknowledge their contribution to the external peer review of your manuscript entitled "Pathogen evasion of social immunity". For those reviewers who give their assent, we will be publishing their names alongside the published article.

Nature Ecology & Evolution offers a Transparent Peer Review option for new original research manuscripts submitted after December 1st, 2019. As part of this initiative, we encourage our authors to support increased transparency into the peer review process by agreeing to have the reviewer comments, author rebuttal letters, and editorial decision letters published as a Supplementary item. When you submit your final files please clearly state in your cover letter whether or not you would like to participate in this initiative. Please note that failure to state your preference will result in delays in accepting your manuscript for publication.

Cover suggestions

As you prepare your final files we encourage you to consider whether you have any images or illustrations that may be appropriate for use on the cover of Nature Ecology & Evolution.

18We accept TIFF, JPEG, PNG or PSD file formats (a layered PSD file would be ideal), and the image should be at least 300ppi resolution (preferably 600-1200 ppi), in CMYK colour mode.

Nature Ecology & Evolution has now transitioned to a unified Rights Collection system which will allow our Author Services team to quickly and easily collect the rights and permissions required to publish your work. Approximately 10 days after your paper is formally accepted, you will receive an email in providing you with a link to complete the grant of rights. If your paper is eligible for Open Access, our Author Services team will also be in touch regarding any additional information that may be required to arrange payment for your article.

Please note that *Nature Ecology & Evolution* is a Transformative Journal (TJ). Authors may publish their research with us through the traditional subscription access route or make their paper immediately open access through payment of an article-processing charge (APC). Authors will not be required to make a final decision about access to their article until it has been accepted. [Find out more about Transformative Journals](https://www.springernature.com/gp/open-research/transformative-journals)

Authors may need to take specific actions to achieve [compliance with funder and institutional open access mandates](https://www.springernature.com/gp/open-research/funding/policy-compliance-faqs). If your research is supported by a funder that requires immediate open access (e.g. according to [Plan S principles](https://www.springernature.com/gp/open-research/plan-s-compliance)) then you should select the gold OA route, and we will direct you to the compliant route where possible. For authors selecting the subscription publication route, the journal's standard licensing terms will need to be accepted, including [self-archiving and license to publish](https://www.nature.com/nature-portfolio/editorial-policies/self-archiving-and-license-to-publish). Those licensing terms will supersede any other terms that the author or any third party may assert apply to any version of the manuscript.

Please use the following link for uploading these materials:
[REDACTED]

19If you have any further questions, please feel free to contact me.

[REDACTED]

Reviewer #1:

Remarks to the Author:

I appreciate this work and the manuscript is well written. The Authors have also satisfactorily replied to questions and expanded the text where needed.

Reviewer #3:

Remarks to the Author:

I appreciate the authors' efforts to revise the manuscript, including the additional experiments related to their findings on ergosterol. At the same time, they decided not to follow my suggestion to include an experiment in which individual fungus strains are subject to selection. I still think that this would have been a useful addition and would have helped clarify some of the current results (see below). At the same time, I understand that these experiments are laborious and require time and effort. The other two reviewers did not seem to be concerned about this, and one could arguably make the case that the study's findings in their current form are of sufficient interest.

Some additional thoughts on the authors' response to my initial comments related to Figure 1: If I understand correctly, the core argument of the authors here is that studying co-infecting strains is crucial because it more closely approximates natural infections, and because they previously found that allogrooming alters the relative abundance of strains, which makes coinfections more interesting. As to the first point, that might be true, but I'm not sure. They cite Balmer & Tanner 2011 in their response, but in the abstract that paper says "multiple-strain infections usually reach considerable frequencies (median 11.3%, mean 21.7% of infections)..." That's not exactly evidence that coinfections are more prevalent than single-strain infections, as claimed by the authors. Rather, coinfections occur and aren't that rare. It also seems that the proportion of coinfections is quite variable across pathogens, so the relevant question here is how frequent natural coinfections of *Metarhizium* are in ants. I couldn't find data on this (admittedly I only did a very cursory search), and my impression is that this is an open question. As to the second point, I agree that the dynamics in mixed-strain infections are interesting. But that doesn't get around the issues that a) it's hard to extrapolate from a single mix and b) you lose some of the information you'd get from single-strain infections. The authors' argument about comparing start and endpoints seems valid, but it would still be good to know what happens with single-strain infections in their experimental design. Maybe something for future work?

Additional comments (mostly minor, but the issue of reduced strain diversity needs to be clarified):

Line 39: omit "(MAMP)"

Line 79: this is the Balmer & Tanner 2011 paper. The authors cite it as showing that "coinfections naturally occur more frequently than single pathogen infections", but I don't think that's a fair representation of that paper (see above).

Line 80: omit comma after "to"

20Line 97: This title seems to contradict the findings in this section: both social and individual immunity reduce the diversity in the pathogen community (rather than maintaining it), and there's in fact no difference between the two treatments in this effect (line 934).

Lines 112-114. This is the one statement that seems to be aligned with the section title, but the authors give no statistical comparison or refer to a table or figure that would show this. I assume they're referring to the data after 5 passages here? But I can't see where this difference is quantified. Lines 153 to 157: I'd suggest a few edits to this sentence so it reads: "Whilst statistical testing is only possible for this prominent R1 strain, the three individual and two social lines in which *M. robertsii* R3 persisted showed similar patterns, with the social lines inducing 75% of the mortality but producing 340% of the spore outgrowth of the individual lines." Also, in the subsequent sentence, I would change "This shows" to "This suggests", simply because you can't do statistical comparisons for R3. Line 167: omit "even"

Lines 206-208: See above – I can't see where the evidence for this is. As far as I can tell, there is no statistical test that would corroborate this conclusion, and the figure legend on line 934 seems to report the contrary. This needs to be clarified.

Author Rebuttal, first revision:

Point to point response to the reviewer's comments.

Reviewer #1:

Remarks to the Author:

I appreciate this work and the manuscript is well written. The Authors have also satisfactorily replied to questions and expanded the text where needed.

We thank the reviewer for this positive response to our revised manuscript.

Reviewer #3:

Remarks to the Author:

I appreciate the authors' efforts to revise the manuscript, including the additional experiments related to their findings on ergosterol. At the same time, they decided not to follow my suggestion to include an experiment in which individual fungus strains are subject to selection. I still think that this would have been a useful addition and would have helped clarify some of the current results (see below). At the same time, I understand that these experiments are laborious and require time and effort. The other two

21reviewers did not seem to be concerned about this, and one could arguably make the case that the study's findings in their current form are of sufficient interest.

We thank the reviewer for appreciating our additional experiments on the chemical pathogen detection, as well as of the relevance of our findings on coinfections in the absence of additional single pathogen strain experiments.

Some additional thoughts on the authors' response to my initial comments related to Figure 1: If I understand correctly, the core argument of the authors here is that studying co-infecting strains is crucial because it more closely approximates natural infections, and because they previously found that allogrooming alters the relative abundance of strains, which makes coinfections more interesting. As to the first point, that might be true, but I'm not sure. They cite Balmer & Tanner 2011 in their response, but in the abstract that paper says "multiple-strain infections usually reach considerable frequencies (median 11.3%, mean 21.7% of infections)..." That's not exactly evidence that coinfections are more prevalent than single-strain infections, as claimed by the authors. Rather, coinfections occur and aren't that rare. It also seems that the proportion of coinfections is quite variable across pathogens, so the relevant question here is how frequent natural coinfections of *Metarhizium* are in ants. I couldn't find data on this (admittedly I only did a very cursory search), and my impression is that this is an open question. As to the second point, I agree that the dynamics in mixed-strain infections are interesting. But that doesn't get around the issues that a) it's hard to extrapolate from a single mix and b) you lose some of the information you'd get from single-strain infections. The authors' argument about comparing start and endpoints seems valid, but it would still be good to know what happens with single-strain infections in their experimental design. Maybe something for future work?

*We thank the reviewer for the careful review and agree that our statement should have been that coinfections occur very frequently in nature, but not that they occur with >50% frequency across host-pathogen systems. We have corrected our sentence accordingly, and also provide an additional reference presenting coinfection rates for a larger diversity of host pathogen systems. Despite the fact that we are also unaware of natural coinfection rates of different *Metarhizium* strains in ants, we still consider them a very relevant selection scenario due to the common natural occurrence of coinfections and the cooccurrence of our chosen strains at high density within a single field population (Steinwender et al. 2014). Moreover, as our main study question was to understand the long-term effects of the ants' modulatory effect during coinfections, our current study approach required a coinfection setup. We do agree with the reviewer that analysing the long-term dynamics and fungal evolution in single infections is a very interesting study question indeed.*

Additional comments (mostly minor, but the issue of reduced strain diversity needs to be clarified):

22Line 39: omit “(MAMP)”

removed

Line 79: this is the Balmer & Tanner 2011 paper. The authors cite it as showing that “coinfections naturally occur more frequently than single pathogen infections”, but I don’t think that’s a fair representation of that paper (see above).

As detailed above, we have toned down our statement.

Line 80: omit comma after “to”

Removed.

Line 97: This title seems to contradict the findings in this section: both social and individual immunity reduce the diversity in the pathogen community (rather than maintaining it), and there’s in fact no difference between the two treatments in this effect (line 934).

Thank you very much for alerting us that we had forgotten to add the statistics that strain diversity is higher in the social immunity treatment in passage 5, whilst there is no more difference at passage 10. We added this information and have rewritten the paragraph to make clearer that the diversity is indeed reduced by both treatments to equal amount after the 10 passages, but that the social immunity treatment led to a slower loss of diversity over the course of the experiment, so that the selection pressure on the successful strains will have differed over the course of the experiment.

Lines 112-114. This is the one statement that seems to be aligned with the section title, but the authors give no statistical comparison or refer to a table or figure that would show this. I assume they’re referring to the data after 5 passages here? But I can’t see where this difference is quantified.

Included now as detailed above.

Lines 153 to 157: I'd suggest a few edits to this sentence so it reads: "Whilst statistical testing is only possible for this prominent R1 strain, the three individual and two social lines in which *M. robertsii* R3 persisted showed similar patterns, with the social lines inducing 75% of the mortality but producing 340% of the spore outgrowth of the individual lines." Also, in the subsequent sentence, I would change "This shows" to "This suggests", simply because you can't do statistical comparisons for R3.

Changed as suggested.

Line 167: omit "even"

Removed.

Lines 206-208: See above – I can't see where the evidence for this is. As far as I can tell, there is no statistical test that would corroborate this conclusion, and the figure legend on line 934 seems to report the contrary. This needs to be clarified.

Included now in the figure legend and the statistics table, as detailed above.

Final Decision Letter:

4th January 2023

Dear Professor Cremer,

We are pleased to inform you that your Article entitled "Pathogen evasion of social immunity", has now been accepted for publication in Nature Ecology & Evolution.

Over the next few weeks, your paper will be copyedited to ensure that it conforms to Nature Ecology and Evolution style. Once your paper is typeset, you will receive an email with a link to choose the appropriate publishing options for your paper and our Author Services team will be in touch regarding any additional information that may be required

24You will not receive your proofs until the publishing agreement has been received through our system

Due to the importance of these deadlines, we ask you please us know now whether you will be difficult to contact over the next month. If this is the case, we ask you provide us with the contact information (email, phone and fax) of someone who will be able to check the proofs on your behalf, and who will be available to address any last-minute problems . Once your paper has been scheduled for online publication, the Nature press office will be in touch to confirm the details.

Acceptance of your manuscript is conditional on all authors' agreement with our publication policies (see www.nature.com/authors/policies/index.html). In particular your manuscript must not be published elsewhere and there must be no announcement of the work to any media outlet until the publication date (the day on which it is uploaded onto our web site).

Please note that *Nature Ecology & Evolution* is a Transformative Journal (TJ). Authors may publish their research with us through the traditional subscription access route or make their paper immediately open access through payment of an article-processing charge (APC). Authors will not be required to make a final decision about access to their article until it has been accepted. [Find out more about Transformative Journals](https://www.springernature.com/gp/open-research/transformative-journals)

Authors may need to take specific actions to achieve [compliance](https://www.springernature.com/gp/open-research/funding/policy-compliance-faqs) with funder and institutional open access mandates. If your research is supported by a funder that requires immediate open access (e.g. according to [Plan S principles](https://www.springernature.com/gp/open-research/plan-s-compliance)) then you should select the gold OA route, and we will direct you to the compliant route where possible. For authors selecting the subscription publication route, the journal's standard licensing terms will need to be accepted, including <https://www.nature.com/nature-portfolio/editorial-policies/self-archiving-and-license-to-publish>. Those licensing terms will supersede any other terms that the author or any third party may assert apply to any version of the manuscript.

An online order form for reprints of your paper is available at <https://www.nature.com/reprints/author-reprints.html>. All co-authors, authors' institutions and authors' funding agencies can order reprints using the form appropriate to their

25geographical region.

We welcome the submission of potential cover material (including a short caption of around 40 words) related to your manuscript; suggestions should be sent to Nature Ecology & Evolution as electronic files (the image should be 300 dpi at 210 x 297 mm in either TIFF or JPEG format). Please note that such pictures should be selected more for their aesthetic appeal than for their scientific content, and that colour images work better than black and white or grayscale images. Please do not try to design a cover with the Nature Ecology & Evolution logo etc., and please do not submit composites of images related to your work. I am sure you will understand that we cannot make any promise as to whether any of your suggestions might be selected for the cover of the journal.

You can generate the link yourself when you receive your article DOI by entering it here: <http://authors.springernature.com/share>.

[REDACTED]

P.S. Click on the following link if you would like to recommend Nature Ecology & Evolution to your librarian <http://www.nature.com/subscriptions/recommend.html#forms>

** Visit the Springer Nature Editorial and Publishing website at http://editorial-jobs.springernature.com?utm_source=ejp_NEcoE_email&utm_medium=ejp_NEcoE_email&utm_campaign=ejp_NEcoE for more information about our career opportunities. If you have any questions please click [here](mailto:editorial.publishing.jobs@springernature.com).**